
# WRF (v4.0)-SUEWS (v2018c) Coupled System: Development, Evaluation and Application

Ting Sun[1,2,*], Hamidreza Omidvar[2,*], Zhenkun Li[3,*], Ning Zhang[4], Wenjuan Huang[3], Simone Kotthaus[5], Helen C. Ward[6], Zhiwen Luo[7], and Sue Grimmond[1]

[1]Department of Meteorology, University of Reading, Reading, UK

[2]Institute for Risk and Disaster Reduction, University College London, London, UK

[3]Shanghai Climate Centre, Shanghai, China

[4]School of Atmospheric Sciences, Nanjing University, Nanjing, China

[5]Institut Pierre Simon Laplace, École Polytechnique, Palaiseau, France

[6]Department of Atmospheric and Cryospheric Sciences, University of Innsbruck, Innsbruck, Austria

[7]Welsh School of Architecture, Cardiff University, Cardiff, UK

[*]These authors contributed equally to this work.

**Correspondence:** Ting Sun (ting.sun@ucl.ac.uk); Sue Grimmond (c.s.grimmond@reading.ac.uk)

**Abstract.** The process of coupling the Surface Urban Energy and Water Scheme (SUEWS) into the Weather research and forecasting (WRF) model is presented, including pre-processing of model parameters to represent spatial variability of surface characteristics. Fluxes and mixed layer height observations in the southern UK are used to evaluate a two-week period in each season. Mean absolute errors, based on all periods, are smaller in residential Swindon than central London for turbulent

sensible and latent heat fluxes ($Q_H$, $Q_E$) with greater skill on clear days at both sites (for incoming and outgoing short- and longwave radiation, $Q_H$ and $Q_E$). Clear seasonality is seen in the model performance: with better absolute skill for $Q_H$ and $Q_E$ in autumn and winter, when there is a higher frequency of clear days, than in spring and summer. As the WRF-modelled incoming shortwave radiation has large errors, we apply a bulk transmissivity derived from local observations to reduce the incoming short-wave radiation input to the land surface scheme - this could correspond to increased presence of aerosols in

cities. We use the coupled WRF-SUEWS system to investigate impacts of the anthropogenic heat flux emissions on boundary layer dynamics by comparing areas with contrasting human activities (central-commercial and residential areas) in Greater London - larger anthropogenic heat emissions not only elevate the mixed layer heights but also lead to a warmer and drier near-surface atmosphere.





# 1 Introduction

Accurate prediction of urban-atmospheric interactions is one essential task of modern numerical weather prediction (NWP) models. There is increasing need to understand city-weather feedbacks and their impact on citizens and infrastructure to facilitate the delivery of Integrated Urban Services (IUS). IUS span weather, climate, hydrometeorological and environmental processes and are related to many urban functions (e.g. urban planning, building design, transport/logistics operation, health, energy infrastructure and operations; Baklanov et al., 2018; Grimmond et al., 2020; Masson et al., 2020).

To improve such predictions, numerous efforts have been made to develop and enhance many urban land surface models (ULSMs, Grimmond et al. 2010a, b; Best and Grimmond 2015), including the SingleLayer Urban Canopy Model (SLUCM, Kusaka et al., 2001), the Building Effect Parameterisation (BEP, Martilli et al., 2002), the Town Energy Balance model (TEB, Masson, 2000), and the Surface Urban Energy and Water Scheme (SUEWS, Järvi et al., 2011). Beyond resolving transfer of energy, water and scalars at land-atmosphere interface, the core tasks of ULSMs are to:

– Characterise the urban surface: the heterogeneous mix of materials and morphology varies from being dominated by built surfaces (i.e. buildings and paved areas) in city centres to having sparse built fractions and more vegetation at more residential outskirts. Morphological variability is driven by the changing heights and spacings of buildings and trees across cities.

– Account for anthropogenic dynamics: As people's behaviour varies, it modifies emissions (e.g. energy, aerosols, wa-
ter) on both regular (e.g. work week and weekends) and irregular (e.g. major sports events such as Olympics, concerts, COVID19) patterns which modifies urban-atmosphere interactions. Thus, the city morphology, or form, remains relatively constant but the functioning of the city varies with changing behaviour patterns.

– Capture the impact of urban-atmosphere interactions on the boundary layer: the urban boundary layer (UBL) is the lowest part of the atmosphere that is directly influenced by the presence of the city. The UBL is characterised by higher wind
speeds, and higher turbulent fluxes than the overlying free atmosphere. Additionally, it features elevated concentrations of pollutants and aerosols, as well as a warmer and more humid near-surface atmosphere.

SUEWS, a widely used and tested ULSM (Table 1), uses a mix of seven land cover types to characterise the surface materials. Anthropogenic heat, water and carbon emissions, with other features (e.g. snow clearing, irrigation) are used to capture behavioural dynamics impacts on urban-atmosphere interactions. Since its development, SUEWS has been regularly enhanced
(e.g. Grimmond et al., 1986; Grimmond and Oke, 1991; Grimmond et al., 1991; Järvi et al., 2011, 2014, 2019; Offerle et al., 2003; Ward et al., 2016; Omidvar et al., 2022) and tested in a wide range of climates and cities worldwide (Table 1). Although operationally simple and scientifically robust, the full SUEWS model has primarily been used offline, preventing many urban-atmosphere feedbacks to be explored with the model. Coupling ULSMs (such as SUEWS) into larger-scale atmospheric models would better represent the land surfaces with more detailed physical processes and is thus expected to enhance the
understanding of urban-atmosphere interactions (Vilà-Guerau de Arellano et al., 2023).



**Table 1.** Recent studies involving SUEWS have undertaken (1) development (**D**) of modules (**M**) and supporting tools (**T**) improvements to coefficients (**C**),(2) applications (**A**) where the model has been evaluated (**E**) or used to assess scenario (**S**) outcome.

| Topic | D | A | City | Reference |
|---|---|---|---|---|
| Application of SUEWS in vegetated areas | T, C | E | Multiple vegetation types | Omidvar et al. (2022) |
| Generation of urban typical meteorological year (uTMY) dataset | M | E, S | London, UK | Tang et al. (2021) |
| Evaluation of storage heat modules | M | E | Basel, Switzerland Heraklion, Greece London, UK | Lindberg et al. (2020) |
| Influence of aerosols on urban water balance | – | S, E | Beijing, China | Kokkonen et al. (2019a) |
| Haze effects on urban water balance | | | Beijing, China | Kokkonen et al. (2019b) |
| SuPy (SUEWS in Python) | T | – | (N/A) | Sun and Grimmond (2019) |
| Impacts of anthropogenic heat and irrigation on surface energy balance | | S, E | Shanghai, China | Ao et al. (2018) |
| $CO_2$ modelling scheme | M | E | Helsinki, Finland | Järvi et al. (2019) |
| Land cover and water use change | – | S | Vancouver, Canada | Kokkonen et al. (2018b) |
| Precipitation effects and reanalysis data | – | S | Vancouver, Canada | Kokkonen et al. (2018a) |
| SUEWS as a core processor of UMEP | T | – | (N/A) | Lindberg et al. (2018) |
| Precipitation intensity impacts on urban climate | | | London, UK | Ward et al. (2017) |
| Comparison with other ULSM | – | – | Singapore | Demuzere et al. (2017) |
| Implications of warming to cold climate cities | | S, E | High latitudes cities | Järvi et al. (2017) |
| Cold climate urban hydrology | – | – | Helsinki, Finland Montreal, Canada Minneapolis, USA Basel, Switzerland | Järvi et al. (2017) |
| Offline evaluation of SUEWS driven by WRF output | – | E | Porto, Portugal | Rafael et al. (2017) |
| Impacts of changes in surface cover, human behaviour & climate on energy partitioning | – | S | London, UK | Ward and Grimmond (2017) |
| Four cities with different climates | | E | Dublin, Ireland Hamburg, Germany Melbourne, Australia Phoenix, USA | Alexander et al. (2016) |
| Radiation flux | – | E | Shanghai, China | Ao et al. (2016) |
| Comparison with other ULSM | – | E | Helsinki, Finland | Karsisto et al. (2015) |
| Evaluation at two UK cities | M, C | E | London, Swindon | Ward et al. (2016) |
| Using Local Climate Zone information as surface characteristics | T | S | Dublin, Ireland | Alexander et al. (2015) |





| Boundary layer modelling and coupling with SUEWS, impacts to heat stress | T | S | Sacramento, USA | Onomura et al. (2015) |
|---|---|---|---|---|
| Snow melt | M | E | Helsinki, Finland | Järvi et al. (2014) |
| SUEWS development | T | E | Montreal, Canada Vancouver, Canada | Järvi et al. (2011) |
| | | | Los Angeles, USA | |
| Impacts of urban design on hydrologic cycle | – | S | Canberra, Australia | Mitchell et al. (2008) |

Here, we couple SUEWS (v2018c; Sun et al., 2019) to the Weather Research and Forecasting (WRF) model (V4.0; Skamarock et al., 2019), an open-source frequently used NWP model. WRF provides the atmospheric forcing to SUEWS, and in turn WRF receives surface-atmosphere feedbacks for the city and the region. In this paper, we describe the structure and key physics of the coupled WRF-SUEWS system (Sect. 2); evaluate WRF-SUEWS at two UK sites (Sect. 3) and explore its

application in modelling dynamics and impacts of anthropogenic heat emissions at the city scale (Sect. 4).

## 2 Development of WRF-SUEWS coupled system

### 2.1 Physical Interactions between WRF and SUEWS

The coupling between WRF and SUEWS occurs via the biophysical interactions between the **land surface** and other physics modules in WRF (Fig. 1):

1. The **radiation** module provides radiative forcing variables, incoming short- $K_\downarrow$ and longwave radiation $(L_\downarrow)$, to the land surface module. The land surface module returns outgoing short- and long-wave radiation ($K_\uparrow$ and $L_\uparrow$).

    2. Atmospheric variables needed for SUEWS, including air temperature $T_a$, relative humidity $RH$, barometric pressure $p_a$ and wind speed $U$ are supplied by the **boundary layer (BL)** module. These are influenced by turbulent transport (i.e., momentum $\tau$, sensible heat $Q_H$ and latent heat $Q_E$ fluxes) from the land surface.

3. Precipitation $P$, is generated by the **microphysics** and **cumulus** modules that parameterise precipitation-related processes at different scales.

### 2.2 Key physics of SUEWS

SUEWS simulates both the energy (Oke, 2002):

$$Q^* + Q_F = Q_H + Q_E + \Delta Q_S \tag{1}$$

and water balances (Grimmond et al., 1986):

$$P + I_e = E + R + \Delta S \tag{2}$$



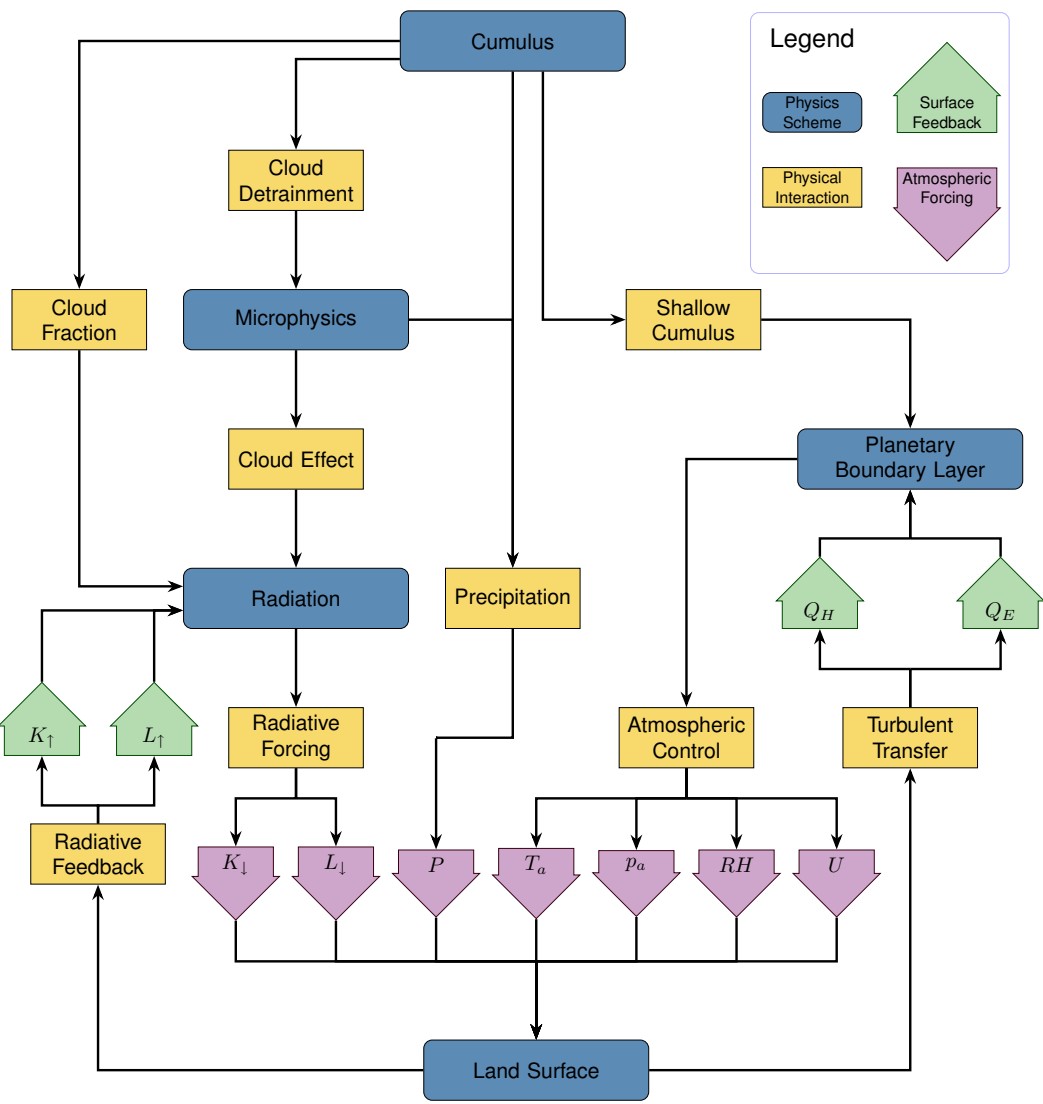

**Figure 1.** Interactions of the five WRF physics (blue) schemes through processes (yellow) and the land surface module variables (purple). Notation defined in Sect. 2.2.

The two are linked through the latent heat $(Q_E)$ or evaporative $E$ fluxes (by the latent heat of vaporisation). The water balance is driven by precipitation $(P)$ and external water use $(I_e)$. Whereas the surface energy balance is driven by the net all-wave radiation $(Q^* = K_\downarrow - K_\uparrow + L_\downarrow - L_\uparrow$ ; $K$ and $L$ denote the short- and long-wave components, respectively; arrows $\downarrow$ and $\uparrow$ in subscript the incoming and outgoing directions, respectively) in all environments but additionally in cities the human activities result in anthropogenic heat flux emissions $(Q_F)$. The turbulent sensible $(Q_H)$, latent $(Q_E)$, net storage heat flux $(\Delta Q_S)$, runoff $(R)$, and change in water storage $(\Delta S)$ each have distinct responses that differ with land use and land cover.





Traditionally, SUEWS has been mostly used for urban areas (Grimmond et al., 1986; Grimmond and Oke, 1991; Järvi et al., 2011; Ward et al., 2016) but for WRF-SUEWS it has been extended to non-urban contexts (Omidvar et al., 2022). Each model grid cell has up to seven land cover types (paved, buildings, deciduous trees, evergreen trees, grass/crops, bare soil and water) whose fractions and properties (e.g. height, albedo, leaf area index) can each vary between grid cells.

In WRF-SUEWS, $Q_F$ is calculated using heating and cooling degree days (HDD and CDD) following the Sailor and Vasireddy (2006) approach:

$$Q_F = \rho_{pop,t} \left[ a_{F0} + a_{F1} CDD + a_{F2} HDD \right] \tag{3}$$

where $a_{F0}, a_{F1}$ and $a_{F2}$ are grid-specific coefficients. The grid population densities $\rho_{pop,t}$ could be a daily mean (i.e. day and night) value (e.g. Ward et al., 2016) or capture the diurnal variations (e.g. Ward and Grimmond, 2017; Ao et al., 2018). Typically, for cities with strong commuting flows (e.g. London), $Q_F$ and $\rho_{pop,t}$ are larger in the central business districts (CBD) during the day due to the commuting, but are higher in residential areas at night. As using daily mean population density may bias $Q_F$ and lose intradaily variability (e.g., large city centre difference between work intensive and non-work periods), following other models (e.g. Allen et al., 2010) we divide the day into four periods (morning transition, day, afternoon transition and night). Day and night population densities are used in their respective periods, and their averages in both transition periods. The other parameter values can be derived from more detailed models that are not rapid enough for NWP (e.g. GQF: Iamarino et al., 2011; Gabey et al., 2018; DASH: Capel-Timms et al., 2020). $\Delta Q_S$ for each grid cell is calculated using the Objective Hysteresis Model (OHM) (Grimmond et al., 1991):

$$\Delta Q_S = \sum_i f_i \left[ a_{1,i} Q^* + a_{2,i} \frac{\partial Q^*}{\partial t} + a_{3,i} \right] \tag{4}$$

where $t$ is the time, $f_i$ is the fraction of area of each land cover type ($i$), and $a_{1-3,i}$ are material-related coefficients for each land cover type that can vary with grid cell (cf. Sect. 3.1). This approach allows much more rapid calculation of this flux than other methods. $Q_E$ is introduced to the Penman-Monteith equation (Grimmond and Oke, 1991):

$$Q_E = \frac{s \left( Q^* + Q_F - \Delta Q_S \right) + \rho c_p V / r_a}{s + \gamma \left( 1 + r_s / r_a \right)} \tag{5}$$

where $s$ is the slope of saturation vapour pressure curve, $\rho$, density of air, $c_p$ specific heat capacity of air at constant pressure, $V$ vapour pressure deficit, $\gamma$ psychrometric 'constant', $r_a$ aerodynamic resistance for heat or water vapour, and $r_s$ surface or canopy resistance. Following Monteith (1965), assuming energy balance closure $Q_H$ is calculated as:

$$Q_H = Q^* + Q_F - \Delta Q_S - Q_E \tag{6}$$

With the latent heat of vaporisation ($L_v$) and Eqn. 5 we obtain $E = \frac{Q_E}{L_v}$ to link surface energy (Eqn. 1) and water (Eqn. 2) balance. $R$ includes the runoff from individual surfaces, in channels and to groundwater. External water use ($I_e$) is estimated based on the automatic and/or manual irrigation or external application (e.g. street cleaning) as follows (Järvi et al., 2011):

$$I_e = A_{irr} \left[ f_{aut} \left( b_{0,a} + b_{1,a} T_d + b_{2,a} t_r \right) + \left( 1 - f_{aut} \left( b_{0,m} + b_{1,m} T_d + b_{2,m} t_r \right) \right) \right] \tag{7}$$





where $A_{irr}$ is the area irrigated, $f_{\text{aut}}$ is the fraction of $A_{irr}$ that is automatically irrigated, $b_{0-2,a,m}$ are site specific coefficients, $T_d$ is the daily mean temperature, and $t_r$ is days since the rain. The net change in the water storage $\Delta S$ (e.g. in soil, water

bodies, on the surface) is determined at each time step as the change of each surface water state compared to the previous time step.

The aerodynamic resistance $(r_a)$ is calculated at first atmospheric level in WRF-SUEWS where the wind speed $(U)$ is determined (Fig. 1):

$$r_a = \frac{\left[\ln\left(\frac{Z_m - z_d}{z_{0m}} - \psi_m\right)\right]\left[\ln\left(\frac{Z_m - z_d}{z_{0v}} - \psi_v\right)\right]}{\kappa^2 u}, \tag{8}$$

where $z_d$ is the zero plane displacement height (m), $z_{0m}$ (and $z_{0v}$) are roughness lengths for the momentum (and heat/water vapour); $u$ is wind speed at height $Z_m$; $\kappa$ is the von-Kármán constant (0.4); and $\psi_m$ (and $\psi_v$) are the atmospheric stability functions for momentum (and water vapour). Stability is determined iteratively using the Obukhov length and initiated with a LUMPS (Grimmond and Oke, 2002) calculated sensible heat flux taking the grid land cover fractions into account.

To compute the grid integrated surface resistance $(r_s)$, its inverse the surface conductance $(g_s)$, is used (Ward et al., 2016):

$$r_s^{-1} = g_s = \sum_i \left( G_{PFT} \left( g_{\text{max},i} f_i \frac{LAI_i}{LAI_{\text{max},i}} \right) g\left(K_\downarrow\right) g(\Delta q) g(T_a) g\left(\Delta\theta_{\text{soil}}\right) \right), \tag{9}$$

The mix of vegetation within the grid is taken into account by considering each vegetation type $i$ with land cover fraction $f_i$, the maximum conductance $g_{\text{max},i}$, leaf area index ($LAI_i$), maximum LAI ($LAI_{\text{max},i}$) and the surface conductance parameter $(G_{PFT})$ determined by plant functional type. Functions $g\left(K_\downarrow\right)$, $g(\Delta q)$, $g(T_a)$, $g(\Delta\theta)$ are related to how the environmental variables – downwelling shortwave $(K_\downarrow)$, specific humidity deficit $(\Delta q)$, air temperature $(T_a)$, and soil moisture deficit

$(\Delta\theta_{\text{soil}})$ – control the surface resistance. These functions have the following forms (Ward et al., 2016):

$$g\left(K_\downarrow\right) = \frac{\frac{K_\downarrow}{G_K + K_\downarrow}}{\frac{K_{\downarrow,\,\text{max}}}{G_K + K_{\downarrow,\,\text{max}}}}, \tag{10}$$

$$g(\Delta q) = G_{q,\,\text{base}} + (1 - G_{q,\,\text{base}}) G_{q,\,\text{shape}}^{\Delta q}, \tag{11}$$

$$g(T_a) = \frac{(T_a - T_L)(T_H - T_a)^{T_c}}{(G_T - T_L)(T_H - G_T)^{T_c}}, \tag{12}$$

$$g\left(\Delta\theta_{\text{soil}}\right) = \frac{1 - \exp\left(G_\theta\left(\Delta\theta_{\text{soil}} - \Delta\theta_{WP}\right)\right)}{1 - \exp\left(-G_\theta\Delta\theta_{WP}\right)}. \tag{13}$$

where the $G$ parameters are related to environmental controls indicated by subscripts: $K$ for solar radiation, $q$ for specific humidity deficit ('base' and 'shape' for base value and curve shape, respectively), $T_a$ for air temperature and $\theta$ for soil moisture deficit. Table 2 gives the values used in the evaluation of the coupled WRF-SUEWS system (detailed in Sect. 3.2). $K_{\downarrow,\text{max}}$ is the maximum incoming shortwave radiation (1200 W m$^{-2}$ used in this work), $T_c = \frac{T_H - G_T}{G_T - T_L}$ with $T_L$ and $T_H$ being the lower and upper limits for switching off evaporation respectively ($T_L = -10°$C and $T_H = 55°$C ); and $\Delta\theta_{WP}$ is the wilting point

(= 120 mm).





LAI varies with growing degree days (GDD) and senescence degree days (SDD), via (Järvi et al., 2011, 2014):

$$LAI_{d,i} = \begin{cases} \min\left(LAI_{\max,i}, (LAI_{d-1,i})^{0.03}GDD \times 5 \times 10^{-4} + LAI_{d-1,i}\right), & T_{\text{BaseSDD}} < T_a < T_{\text{BaseGDD}} \\ \max\left(LAI_{\min,i}, (LAI_{d-1,i})^{0.03}SDD \times 5 \times 10^{-4} + LAI_{d-1,i}\right), & T_{\text{BaseGDD}} < T_a < T_{\text{BaseSDD}} \end{cases} \tag{14}$$

where the previous day (subscript $d-1$) LAI is used with the base temperature corresponding to the initiation of leaf-off ($T_{\text{BaseSDD}}$) and leaf-on periods ($T_{\text{BaseGDD}}$). The model also requires $LAI_{\max,i}$ and $LAI_{\max,i}$ for each vegetation type $i$.

SUEWS accounts for the running water balance of the multiple surface types. The water amount on the canopy of each surface ($C_i$) (Grimmond et al., 1991) determines the surface resistance between dry and wet ($r_s = 0\,\text{s}\,\text{m}^{-1}$) by replacing $r_s$ in Eqn. 5 with $r_{ss}$ (Shuttleworth, 1978):

$$r_{\text{ss}} = \left[\frac{W}{r_b(s/\gamma + 1)} + \frac{(1-W)}{r_s + r_b(s/\gamma + 1)}\right]^{-1} - r_b(s/\gamma + 1) \tag{15}$$

where $W$ is a function of the relative amount of water present on each surface to its water storage capacity $S_i$:

$$W = \begin{cases} 1 & C_i \geq S_i \\ \frac{K_r - 1}{K_r - S_i/C_i} & C_i < S_i \end{cases} \tag{16}$$

$K_r$ depends on the aerodynamic and surface resistances:

$$K_r = \frac{(r_s/r_a)/(r_a - r_b)}{r_s + r_b(s/\gamma + 1)} \tag{17}$$

where $r_b$, the boundary layer resistance, is a function of friction velocity $u_*$ (Shuttleworth, 1983):

$$r_b = 1.1u_*^{-1} + 5.6u_*^{\frac{1}{3}} \tag{18}$$

Eqns. 15 to 18 ensure that the surface resistance $r_{ss}$ has a smooth transition from $0\,\text{s}\,\text{m}^{-1}$ (a completely wet surface) to $r_s$ (a dry surface).

## 2.3 Major updates since SUEWS v2018c

This work presents a coupling framework and its evaluation using SUEWS v2018, ensuring consistency in internal physics with the offline version for the comparison later in Sect. 3.4. However we note that the coupling structure designed for WRF-
SUEWS enables seamless upgrades to more recent SUEWS versions.

Current offline versions of SUEWS have options not in the coupled WRF-SUEWS system, including:

- *CO$_2$ fluxes*: local scale anthropogenic and biogenic urban-atmosphere exchanges (Järvi et al., 2019).

- *Roughness sub-layer profiles*: diagnosis of air temperature, humidity, and wind speed within the roughness sub-layer (Theeuwes et al., 2019; Tang et al., 2021).

- *2-D radiation profiles*: solar and thermal-infrared radiation for multi-layer urban canopies (Hogan, 2019).

- *ESTM*: heat storage estimation using surface temperature and thermal properties (Lindberg et al., 2020).



## 2.4 Technical implementation of the WRF-SUEWS coupling

The following are considered in the design of coupled WRF-SUEWS system:

- *Performance*: Given coupling with file-based IO (input-output) exchanges has unacceptable computational performance, we use the `SuMin` module under the WRF framework (Figs. 2 and 3).

- *Extendibility*: As SUEWS is regularly enhanced (Table 1), it is desirable or even essential, for the coupled system to use the full capacity of the standalone SUEWS.

- *Sustainability*: Given the vast community effort to build and improve sophisticated software systems coupling should not be limited to one version. Instead a highly standardised coupling procedure is required to be sustainable (Meyer et al.,
165 2020).

To address these, the coupled WRF-SUEWS system uses an adaptive intermediate layer `SuMin` (**SUEWS** in a **Min**imum mode) to link both models (Fig. 5). From SUEWS, `SuMin` calls the main SUEWS calculator to conduct all core SUEWS physics calculations. Whereas from WRF, `SuMin` is linked to the `module_sf_suews` via `suews_1d` as a complete land surface model that can be used by WRF dynamics solver (i.e., ARW solver) via the surface driver (Fig. 5). By coupling SUEWS
and WRF this way, fast prototyping of new functionalities is possible on the SUEWS side while maintaining a stable coupling to the more complex WRF. When new SUEWS features are available to be fully coupled, appropriate switches can be activated to incorporate them within the whole WRF system. This intermediate-layer-based approach allows efficient communication between SUEWS and other models (e.g., SuPy, Sun and Grimmond, 2019) through an explicit, unified interface. Thus, SUEWS can be potentially coupled to other weather/climate modelling systems (e.g., openIFS, ECMWF, 2021).

A Python-based WRF-SUEWS pre-processor system (WSPS, Fig. 5) formats the data to allow the additional parameters not in standard WRF input files (e.g. input variables in `wrfinput.nc` , `namelist` based configurations; cf. IO workflow in Fig. 3) to be incorporated, rather than modifying the WRF Preprocessing System (WPS). The WSPS can be used for offline model spin-up runs to obtain the appropriate required initial conditions for WRF-SUEWS. The files prepared are:

i `wrfinput.nc`: a modified version of WRF inputs with initial model states and other static properties; and

ii `namelist.suews`: the global configuration for SUEWS model.

To generate these, four types of inputs are needed (Fig. 3):

i1 Standard WRF input files for WPS: The geographic and meteorological data are processed by WPS to produce `wrfinput.nc` files for the model domains and provide the template for the WSPS.

i2 Additional input files: The static SUEWS-specific properties (e.g. land cover, population density, building morphology)
and optional files (e.g. suitable default parameters to be used when known values are unavailable), which will precede the same information, if available, in the standard WRF input files within the coupled WRF-SUEWS system. Note in the London context this is not required (see later in Sect. 3).





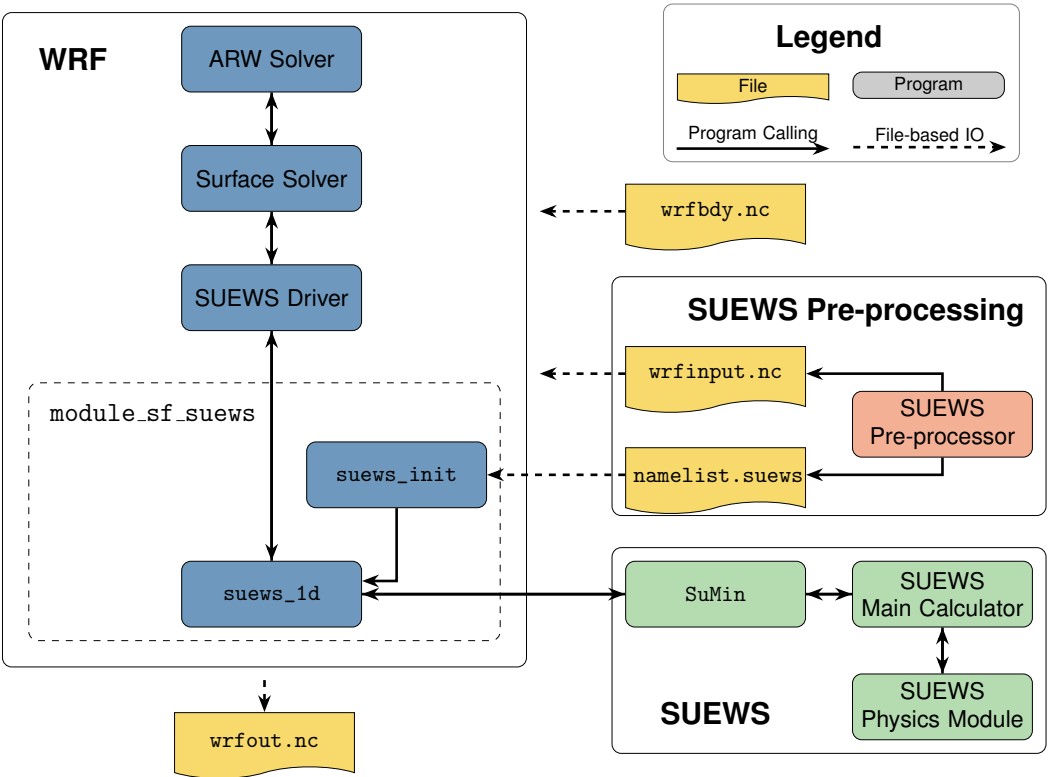

**Figure 2.** WRF-SUEWS system consists of the new |module_sf_suews| added into WRF (blue) to interact with SUEWS via |SuMin| (green) using input files pre-processed by SUEWS pre-processor (yellow). File-based input/output flow (dashed) and runtime calling logic (solid lines) are shown.

i3 Standard SUEWS input files: Files used by SuPy (Sun and Grimmond, 2019) for offline spin-up simulations to obtain appropriate model initial conditions (an example shown in Sect. 3.2). The SUEWS settings (e.g. physics options, 190 population density profiles) are used to create the namelist.suews global settings.

i4 Land cover reclassification settings: In namelist.suews the relations between land covers for WRF and SUEWS (Sect. 2.4) are prescribed.

The WSPS input files (Fig. 3) can have different spatial resolutions between files. The implemented netCDF processor obtains the static properties (i2) and initial condition (i3); and resamples them to the geospatial configuration (projection 195 method, resolution and averaging strategy) of the base wrfinput.nc (i1), to produce the wrfinput.nc files for WRF-SUEWS. Subsequently, the namelist.suews can be easily modified by hand without going through the WSPS if useful (e.g., to test different configurations for spin-up, or change land cover mapping relations).



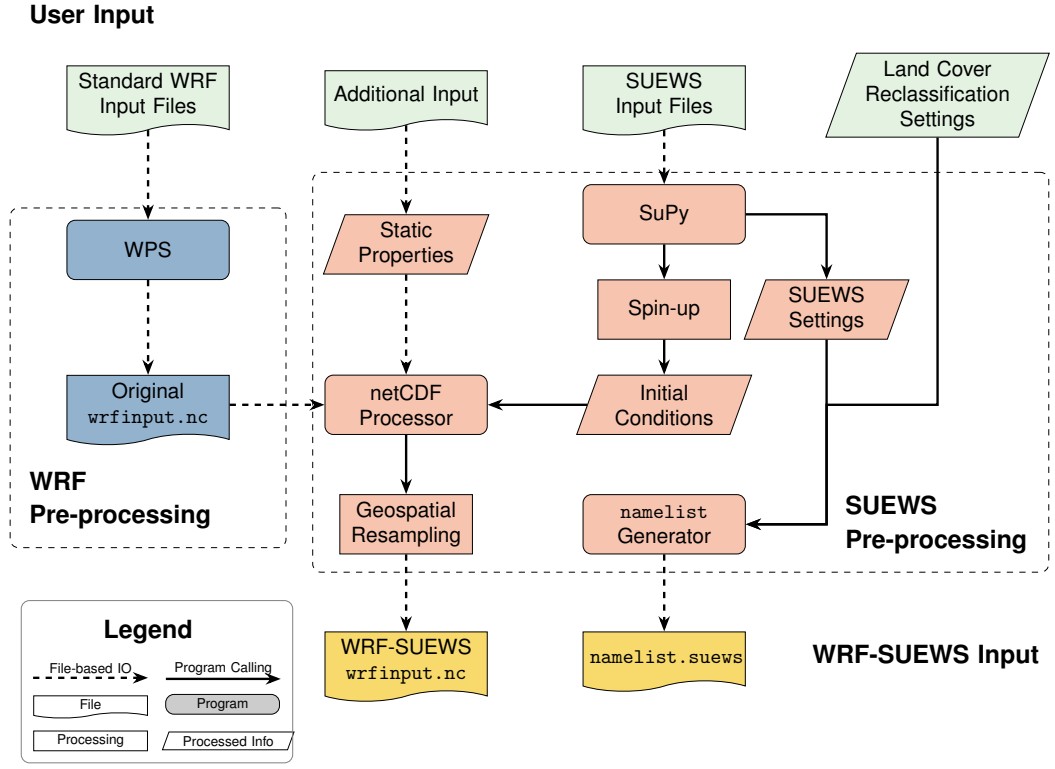

**Figure 3.** Workflow for WRF-SUEWS pre-processing system (WSPS)

The seven land cover (LC) types can be assigned different parameter values (Table 2) per grid cell and/or can change with time. For example, the 'grass' vegetation type can have varying parameters with season (e.g. rice-wheat rotation). The WSPS uses the `namelist.suews` global configuration file to translate the WRF land use (LU) data (e.g. IGBP (International Geosphere-Biosphere Programme)-Modified MODIS (MODerate resolution Imaging Spectro-radiometer) with 20 LU classes, or the USGS with 24 LU classes (NCEP, 2000), to the seven LC classes (Fig. 4). Each SUEWS LC may combine fractions from multiple IGBP LU. Through this reclassification, WRF-SUEWS can use existing LU data for SUEWS simulations while allowing the other parameters to vary between grids. Note a flexible number of WRF LUs (up to 100 in this release) can be specified to compose a SUEWS LC so an extremely heterogeneous LU composition can be accounted for.

## 2.5 Bulk transmissivity based solar radiation correction

Incoming shortwave radiation $(K_\downarrow)$ is known to be overestimated by WRF because of unresolved clouds and/or aerosols (Jimenez et al., 2016; Lapo et al., 2017). However, if the forcing radiation is too large the other surface fluxes and variables will be impacted. Thus, they should not compare well to observations or alternatively if they do compare well, the variables are correct for the wrong reason. In urban areas, even on clear days, there is often a large presence of aerosols that impact





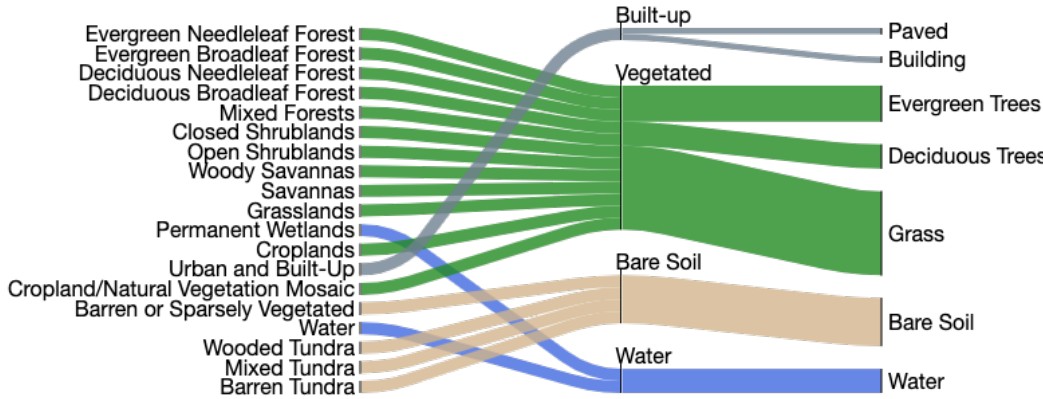

**Figure 4.** The WSPS can be used to reclassify the WRF/IGBP default MODIS 20-category land uses to SUEWS specific land covers: specifically, the 20 MODIS land use categories (left) with four general categories (middle) are reclassified into 7 SUEWS land covers (right).

bulk transmissivities (e.g. Shanghai: Xu et al., 2011; Ao et al., 2018; Beijing: Dou et al., 2019; Sun et al., 2022; London: Ryder and Toumi, 2011; Kotthaus and Grimmond, 2018a; Warren et al., 2018). A bulk atmospheric transmissivity ($\tau$) can be specified in the `namelist.suews` to partially correct the overestimation of $K_\downarrow$ by WRF, which can be determined using $K_\downarrow$ at both the top of atmosphere ($K_{\downarrow,TOA}$) and the surface ($K_{\downarrow,S}$) are used (Oke, 2002):

$$\tau = \frac{K_{\downarrow,s}}{K_{\downarrow,TOA}} \tag{19}$$

As $\tau$ can vary seasonally (e.g. the cases in London and Swindon as shown in Table 4) we determine the median clear sky difference ($\Delta\tau$) between $\tau_{\mathrm{WREE}}$ and $\tau_{\mathrm{abs}}$ from analysis of clear sky days observations around the peak $K_\downarrow$ (which occurs between $40\%$ and $60\%$ of the daylight hours). The $K_\downarrow$ forcing (Fig. 1) for SUEWS in WRF is calculated as:

$$K_{\downarrow,W-S} = K_{\downarrow,WRF} - (1 - \Delta\tau_c) K_{\downarrow,TOA} \tag{20}$$

Given the empirical nature of the parameter values, this correction can only be applied where observations are available. Here, we apply the correction to all time periods but separate the evaluation(Sect. 3.4.1) by sky conditions to assess effectiveness. Obviously, this simple correction is not a complete solution but rather an attempt to obtain more accurate $K_\downarrow$ forcing for the coupled SUEWS, and hence, better surface feedbacks for the WRF atmospheric modules (Fig. 1).

# 3 Evaluation of WRF-SUEWS at two UK urban sites

## 3.1 Surface characteristics of evaluation sites

WRF-SUEWS is evaluated at the same two UK sites as in a previous SUEWS evaluation study (Ward et al., 2016; W16 hereafter) for consistency - these sites exhibit distinct urban characteristics:





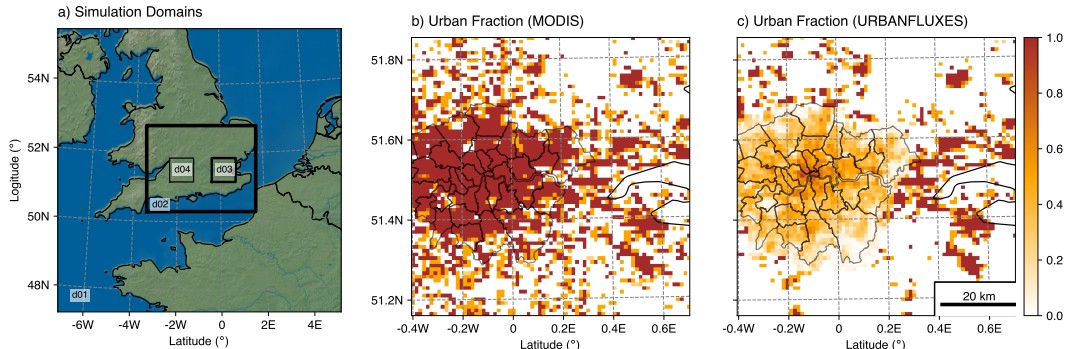

**Figure 5.** Model domain configurations: (a) four simulation domains (d01-d04) and urban land cover (paved and buildings) fraction in d03 (1 km resolution) based on (b) the WRF default MODIS dataset and (c) updated information for Greater London from URBANFLUXES project (Lindberg et al. 2020). The land cover information is accessible in Sun et al. (2023a).

- **KCL**: King's College London Strand Street campus ($51°30'$ N $0°07'$ W): a dense central business district area in London (d03 of Fig. 5), and

- **SWD**: Swindon ($51°35'$ N $1°48'$ W): a residential area in the town of Swindon (d04 of Fig. 5).

WRF-SUEWS is set up with four nested model domains (Fig. 5) with grid spacing (domain and number of grids) being $9\,\mathrm{km}(\,\mathrm{d}01, 100 \times 100), 3\,\mathrm{km}(\,\mathrm{d}02, 115 \times 91)$ and $1\,\mathrm{km}$ (d03 and d04, $76 \times 76$ )

As the default MODIS-based built (building and paved) fraction does not capture the surface heterogeneity within Greater London (Fig. 5b), we replace it with a high resolution (i.e. $2.5\,\mathrm{m}$ ) land cover map (Fig. 5c) derived from an earth observation

(EO) VHR SPOT imagery (Mitraka et al., 2016; Marconcini et al., 2017) with more realistic surface information. This high resolution dataset is processed using UMEP (Lindberg et al., 2018) to derive both land cover fractions for the seven SUEWS classes and other morphological parameters of roughness elements (e.g. building and vegetation heights, frontal area index; Table 2). The resulting dataset is upscaled to obtain $1\,\mathrm{km}$ resolution data for d03. As the equivalent detailed land cover information are unavailable for d04, the land cover (plus building and vegetation height) for the single grid where Swindon

site is located is modified based on values in Ward et al. (2016) (Table 2).





Geoscientific Model Development Discussions — Open Access EGU

**Table 2.** Key parameters assigned in the four model domains (Fig. 5) vary with land cover that is impervious (paved: PAV, buildings: BDG) and pervious (deciduous trees and shrubs: DCT and evergreen: EVT, grass or crops: GRA or CRP, bare soil: BSO and water: WAT). Anthropogenic heat flux coefficients vary between weekday (WD) and weekend (WE). Data sources are: Chrysoulakis et al. (2018):C18, Lindberg et al. (2020): L19, Omidvar et al. (2022): O22, and Ward et al. (2016): W16.

| Parameter | Units | PAV | BDG | DCT | EVT | GRA | BSO | WAT | Densely built-up [a] | Suburban [b] | Natural [c] |
|---|---|---|---|---|---|---|---|---|---|---|---|
| *Interception capacity (Source: W16; Eqn. 16)* | | | | | | | | | | | |
| $S_i$ | mm | 0.48 | 0.25 | 1.3 | 0.8 | 1.9 | 1.9 | - | - | - | - |
| *Phenology (Source: O22; Eqn. 14)* | | | | | | | | | | | |
| $LAI_{min}$ | m² m⁻² | - | - | 0.66 | 0.56 | 0.35 | - | - | - | - | - |
| $LAI_{max}$ | m² m⁻² | - | - | 2.9 | 2.46 | 2.15 | - | - | - | - | - |
| $T_{Base,SDD}$ | °C | - | - | 7.3 | 4 | 4 | - | - | - | - | - |
| $T_{Base,GDD}$ | °C | - | - | 20.6 | 14 | 16.5 | - | - | - | - | - |
| *Albedo (Sources: W16, O22)* | | | | | | | | | | | |
| $\alpha_{LAI,min}$ | - | 0.11 | 0.11 | 0.10 | 0.09 | 0.16 | 0.21 | 0.12 | - | - | - |
| $\alpha_{LAI,max}$ | - | - | - | 0.13 | 0.11 | 0.19 | - | - | - | - | - |
| *OHM (Source: W16; Eqn. 4)* | | | | | | | | | | | |
| $a_1$ | - | 0.3 | 0.337 | 0.215 | 0.215 | 0.215 | 0.335 | 0.5 | - | - | - |
| $a_2$ | h | - | - | 0.325 | 0.325 | 0.325 | 0.335 | 0.21 | - | - | - |
| $a_3$ | W m⁻² | -42.4 | -33.9 | -19.9 | -19.9 | -19.9 | -35.28 | -39.1 | - | - | - |
| *Surface conductance (Sources: W16, O22; Eqns. 9 to 13)* | | | | | | | | | | | |
| $G_{max}$ | m s⁻¹ | - | - | 21.2 | 20.5 | 38.6 | - | - | - | - | - |
| $G_{LAI}$ | - | - | - | 1 | 1 | 1 | - | - | - | 3.5 | - |
| $G_K$ | W m⁻² | - | - | 100 | 62 | 87 | 108.93 | - | - | - | 200 |
| $G_{q,base}$ | - | - | - | 0.44 | 0.39 | 0.47 | 0.93 | - | - | - | 0.13 |
| $G_{q,shape}$ | - | - | - | 0.9 | 0.9 | 0.9 | 0.96 | - | - | - | 0.7 |
| $G_T$ | °C | - | - | 30 | 30 | 30 | 42.26 | - | - | - | 30 |
| $G_\theta$ | mm⁻¹ | - | - | 0.028 | 0.022 | 0.022 | 0.041 | - | - | - | 0.05 |
| *Anthropogenic heat (WD/WE) (Source: W16,Eqn. 3)* | | | | | | | | | | | |
| $a_{F0}$ | W m⁻² (inh ha⁻¹)⁻¹ | - | - | - | - | - | - | - | 0.37/0.34 | 0.14/0.13 | - |
| $a_{F1}$ | W m⁻² (inh ha⁻¹)⁻¹ | - | - | - | - | - | - | - | 0/0 | 0/0 | - |

| $a_{F2} \times 100$ | W m$^{-2}$ (inh ha$^{-1}$)$^{-1}$ | - | - | - | - | - | - | 0.73/0.67 | 0.37/0.38 | - |

[a] $0.6 \leq f_{PAV} + f_{BDG} \leq 1$

[b] $0.16 \leq f_{PAV} + f_{BDG} < 0.6$

[c] $f_{PAV} + f_{BDG} < 0.16$





To help assign SUEWS parameters related to the surface characteristics, the land cover characteristics of the 1362 d03 grid cells within the Greater London area are analysed by plan area fractions of paved ($f_{\text{PAV}}$) and building ($f_{\text{BDG}}$) land covers (Table 2). The most common ( N = 171), LC grid combination (i.e., $f_{\text{PAV}} - f_{\text{PAV}}$ 0.05 fraction bins) is predominately pervious (notably grass) with minimal impervious area ($< 0.05$ for both paved $f_{\text{PAV}}$ and buildings $f_{\text{PAV}}$. The second most frequent (N =

112, $f_{\text{PAV}} = 0.15$ and $f_{BDG} = 0.1$) is also largely pervious. It is also noting that KCL and SWD (blue dots in Table 2) reside in densely built-up and moderately pervious domains, respectively, indicating the different nature in land cover composition. Because high resolution property information is not readily available across the evaluation domains, the surface related SUEWS parameters (e.g. albedo, emissivity, OHM coefficients, etc.) are simplified into three classes based on paved and buildings fraction ($f_{\text{PAV+BDG}}$) from the gridded land cover (Table 2): (a) densely built areas ($f_{\text{PAV+BDG}} > 0.6$) are assigned parameter

values of KCL (W16), (b) suburban areas ($0.16 < f_{\text{PAV+BDG}} \leq 0.6$) are assigned parameter values of SWD (W16); (c) natural surfaces ($0 < f_{\text{PAV+BDG}} \leq 0.16$ ) are assigned based on parameter values of dominant vegetation (Omidvar et al., 2022). In doing so, we can utilise the available property data to accurately represent surface heterogeneity. Note, the WRF-SUEWS system allows grid cell-level surface characteristic parameters assignment (e.g. SUEWS simulation of the Greater London by Lindberg et al., 2020).

Given the importance of population density to anthropogenic heat emissions (Eqn. 3), output area day and night-time population data (UK ONS, 2013) are resampled to 1 km resolution for d03. For the grid point of SWD in d04, the Ward et al. (2016) values are used; otherwise, zero anthropogenic heat emission is set with population density being zero.

### 3.2   Model setup and spin-up

WRF-SUEWS is run with two-way nesting mode of 33 vertical levels (top at 5 kPa 11 layers in the boundary layer below
2000 m with lowest levels in d03 and d04 being $\sim 40$ m agl) for all four domains (Fig. 5). We note more vertical levels may be needed in detailed investigations of atmospheric features (e.g. temperature, precipitable water, etc.); here a moderate number of vertical levels are used as a balance between the computational cost and necessary representation of atmospheric profiles considering the focus of this work on the model development and evaluation of essential urban-atmospheric interactions. The atmospheric boundary conditions used are the $1° \times 1°$ (latitude $\times$ longitude) National Centre for Environmental Prediction
FNL data (NCEP, 2000). The well-tested WRF 'CONUS' physics suite (configuration since v3.9) is used with the land surface scheme changed to SUEWS (Table 3). The SUEWS physics schemes (Table 3, details provided in Sun et al., 2019) are selected for simplicity including using the building and tree heights with rule of thumb method (Grimmond and Oke, 1999) for momentum roughness length and displacement height; additionally, snow and irrigation modules are turned off (following W16's KCL and SWD configuration). We use the WRF adaptive time step option to reduce the total run time while being numerically
stable considering both the horizontal and vertical extent (Hutchinson, 2007). For us, the adopted time steps for domain 1 to 4 are around 72, 9, 3 and 3 seconds.



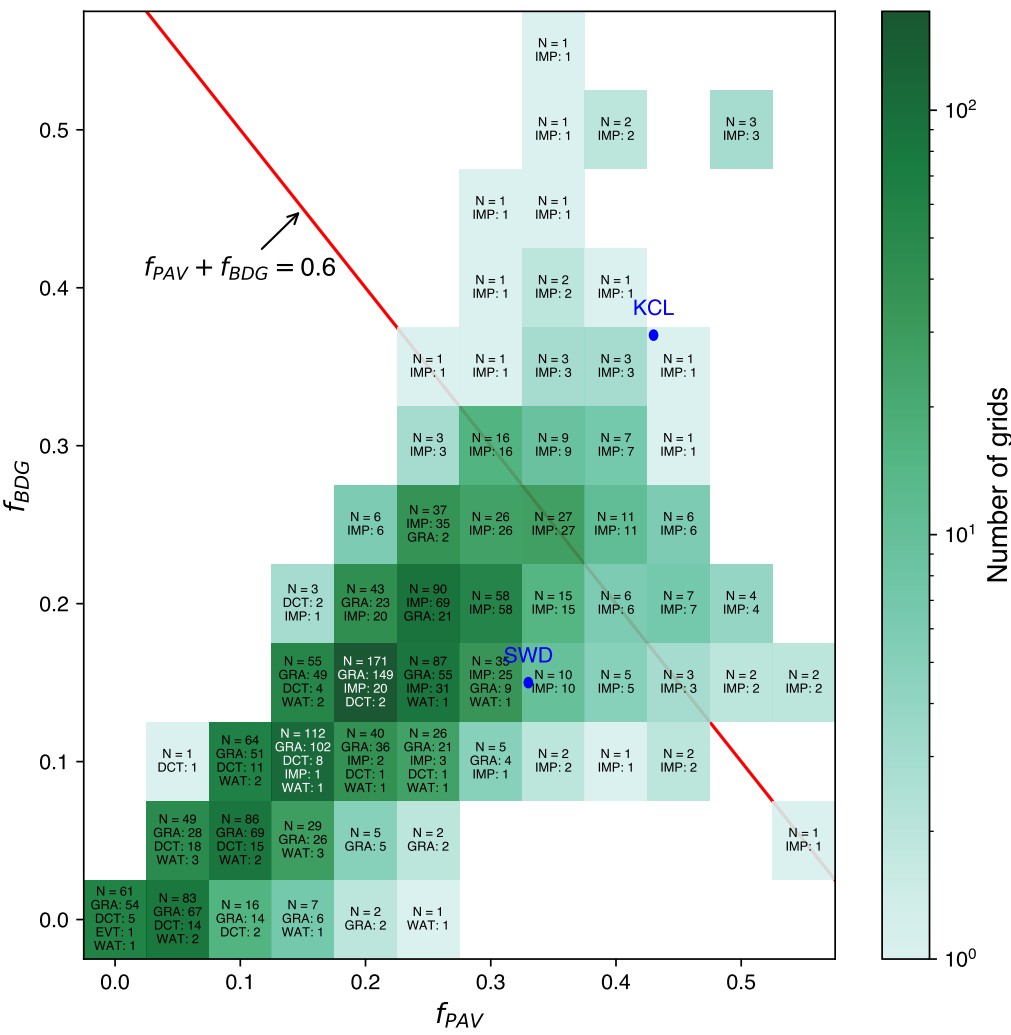

**Figure 6.** Frequency (colour, N) of land cover characteristics in d03 (Fig. 5) for the Greater London (GL) area (1362 grids of 1 km$^2$). The individual grid cells are categorised first by the greatest land cover fraction with impervious split between paved surfaces (PAV) and buildings (BDG). Other fractions are: deciduous trees (DCT), evergreen trees (EVT), grass (GRA), bare soil (BSV), and water (WAT). The blue dots indicate the cover around the KCL and SWD evaluation sites.

**Table 3.** Physics scheme in WRF and SUEWS option tested for use in coupled simulations. The local (internal) option number is given in column #.

| Name in setup | Description | # | Scheme | References |
|---|---|---|---|---|
| *WRF* | | | | |
| mp_phusics | Micro-physics | 18 | Thompson | Thompson et al. (2008) |





| `cu_physics` | Cumulus parametrisation | 6 | Tiedtke | Tiedtke (1989); Zhang et al. (2011) |
|---|---|---|---|---|
| `ra_lw_physics` | Longwave | 4 | RRTMG | Iacono et al. (2008) |
| `ra_sw_phusics` | Shortwave | 4 | RRTMG | Iacono et al. (2008) |
| `bl_pbl_physics` | Planetary boundary layer | 2 | Mellor-Yamada-Janiic | Janjić (1994) |
| `sf_sfclay_physics` | Surface layer | 2 | Eta similarity | Janjić (1994) |
| `sf_surface_physics` | Land Surface Model | 9 | SUEWS | Grimmond et al. (1986); Grimmond and Oke (1991); Järvi et al. (2011); Ward et al. (2016) |
| *SUEWS* | | | | |
| `RoughLenHeatMethod` | Roughness length for heat | 2 | - | Kawai et al. (2009) |
| `RoughLenMomMethod` | Roughness length for momentum | 2 | - | Grimmond and Oke (1999) |
| `StabilityMethod` | Stability function | 3 | | Dyer (1974); Van Ulden and Holtslag (1985); Högström (1988) |
| `EmissionsMethod` | Anthropogenic heat | 2 | - | Järvi et al. (2011) |
| `NetRadiationMethod` | Radiation components | 1 | - | Loridan et al. (2011) |
| `StorageHeatMethod` | Storage heat flux | 1 | - | Grimmond and Oke (1991) |
| `SnowUse` | Snow calculation | 0 | - | Järvi et al. (2014) |

We evaluate WRF-SUEWS during the four seasons using two-week periods in 2012 (Table 4). To generate appropriate initial conditions (i.e., model spin-up), we conduct offline SUEWS runs driven by observations collected at KCL and SWD (Table 2) for 2012 until the soil moisture converges ($< 0.1\%$ difference in last time step between consecutive years): 15 years are needed

for KCL while 5 years for SWD. For fully vegetated grids (i.e. $f_{\text{DCT/EVT/GRA}} = 1$), the observations at SWD are used as forcing conditions to spin-up SUEWS for 4 years before convergence. The required initial states (i.e. soil moisture, leaf area index) are used for each WRF-SUEWS period (Table 4). For grid cells with $f_{\text{PAV+BDG}} > 0.6$, the KCL-based initial states are prescribed to represent areas dominated by impervious surfaces; while SWD is used for the rest that are not completely vegetated. The appropriate complete pervious cover type values are assigned to the pervious cells based their dominant land cover type.





**Table 4.** Time periods in 2012 used to evaluate the WRF-SUEWS (Sect. 3.2) in London (KCL) and Swindon (SWD) with the observed air temperature (KCL: 49.6 m agl, SWD: 10.6 m agl) and rainfall. Measurement details given in Ward et al. (2013); Kotthaus and Grimmond (2014a, b). Note day light savings impacts all except for the January period, i.e. people's activities (e.g. work times) are one hour earlier than UTC. $\Delta\tau_c$ is median clear sky transmissivity difference between $\tau_{WRF}$ and $\tau_{obs}$. The clear-sky days are determined with mean $\tau_{obs} > 0.8$.

| Period | Daily mean temperature ($^{\circ}$C) | | Total rainfall (mm) | | Number of clear days | | $\Delta\tau_c$ | |
|---|---|---|---|---|---|---|---|---|
| | KCL | SWD | KCL | SWD | KCL | SWD | KCL | SWD |
| Jan 16-30 | 6.1 | 5.2 | 14.8 | 55.8 | 5 | 5 | 0.14 | 0.11 |
| Apr 11-25 | 8.3 | 6.6 | 42.4 | 67.8 | 2 | 2 | 0.22 | 0.20 |
| Jul 16-30 | 18.4 | 17.0 | 9.6 | 11.0 | 3 | 2 | 0.07 | 0.00 |
| Oct 1-14 | 11.5 | 9.9 | 42.6 | 19.0 | 5 | 3 | 0.16 | 0.05 |

## 3.3 Evaluation data and metrics

The W16 evaluation of SUEWS (v2016a) uses 60-min radiation and turbulent fluxes observed at KCL and SWD (Ward et al., 2013; Kotthaus and Grimmond, 2014a, b). Both flux towers are located close to the centre (within a $200-300$ m radius circle) of a $1$ km$^2$ model grid cell. The source areas of the observed turbulent fluxes have the probable $50\%$ contribution from within $\sim 400$ m of the flux tower at KCL (Kotthaus and Grimmond, 2014b) and the probable $80\%$ contribution from within $\sim 700$ m at SWD (Ward et al. 2013). Here, we average the 30-min sample output from land surface scheme (i.e., SUEWS) to $60$ min to compare with the observations.

The mixed layer height (MLH), derived from attenuated backscatter observed with a Vaisala CL31 ceilometer at Marylebone Road (MR) in London (Kotthaus et al., 2016; Kotthaus and Grimmond, 2018a, b), is used to evaluate the model's ability to predict atmospheric boundary layer (ABL) dynamics. Various observations can be used to obtain the height of the boundary layer but the results depend on the variable used (Kotthaus et al., 2023). For example, differences occur between using temperature inversion and MLH (e.g., at night, Kotthaus and Grimmond, 2018a), or between MLH and the turbulence-derived mixing height (MH, Kotthaus et al., 2018) - the height where the vertical velocity variations falls below a threshold (Barlow et al., 2011; Halios and Barlow, 2017).

On the other hand, when using the Mellor-Yamada-Janjic scheme, the WRF output PBLH (planetary boundary layer height) is derived from the height where turbulent kinetic energy falls below $0.2$ m$^2$ s$^{-2}$ (Banks et al., 2016; Janjić, 1994).

The comparison of the aerosol-derived MLH from observations and the turbulence-based mixing height diagnosed from the model output (WRF PBL, hereafter referred to as WRF MH) may be affected to systematic differences, including those associated with vertical resolution (Kotthaus et al., 2023).

The evaluation metrics used with the number of data points ($N$) available from the model output ($Y_{mod}$) and observations ($Y_{obs}$) time series are:

1. hit rate (HR):





$$\mathrm{HR} = \frac{\sum_{j=1}^{N} H\left(\delta_{Y,j} - |Y_{\mathrm{mod},j} - Y_{\mathrm{obs},j}|\right)}{N} \tag{21}$$

with Heaviside step function $H$ defined by

$$H(x) = \begin{cases} 0, & x < 0 \\ 1, & x \geq 0 \end{cases} \tag{22}$$

305 and the threshold $\delta_{Y,j}$ being a value dependent on evaluation variable $Y$. We use the HR to evaluate the surface energy fluxes with $\delta_{Yj} = 50$ W m$^{-2}$ for radiative ($K_\downarrow, K_\uparrow, L_\downarrow, L_\uparrow, Q^*$) and $\delta_{Y,j} = 0.1Q^* + 50$ W m$^{-2}$ for turbulent ($Q_E$ and $Q_H$) fluxes (Hollinger and Richardson, 2005), respectively. If HR $= 0$, it suggests none of model predictions are within the acceptable threshold set; while HR $= 1$ indicates all fall into the acceptance range.

2. mean absolute error (MAE):

310
$$\mathrm{MAE} = \frac{\sum_{j=1}^{N} |Y_{\mathrm{mod}} - Y_{\mathrm{obs}}|}{N} \tag{23}$$

3. mean bias error (MBE):

$$\mathrm{MBE} = \frac{\sum_{j=1}^{N} (Y_{mod} - Y_{obs})}{N} \tag{24}$$

Both MAE and MBE have units of the variable analysed (i.e. W$^{-2}$ for fluxes, m for MLH or MH) with an ideal value of 0 indicating perfect agreement with the observations. The MAE, unlike the root mean square error, treats all error equally 315 (Willmott et al., 2017).

### 3.4 Evaluation results

#### 3.4.1 Effect of bulk transmissivity correction on solar radiation

First, we evaluate WRF-SUEWS' skill at predicting incoming shortwave radiation ($K_\downarrow$) as it is crucial to driving surface-atmosphere processes (Fig. 1). Given fixing WRF's RRTMG radiation scheme (Iacono et al., 2008) tendency to overestimate 320 $K_\downarrow$ is beyond the scope of this study, we modify $K_\downarrow$ for each grid to ensure that the land surface receives the appropriate energy to drive the land surface scheme (e.g. SUEWS) based on the bulk transmissivities differences on clear sky days (Fig. 7; correction methodology in Sect. 2.5).

Generally, after the correction is applied the modified $K_\downarrow$ agrees better with observations on clear sky days (Fig. 8) with reduced overestimation. The improvement in HR is minimal, but MAE and MBE become smaller. This type of correction may



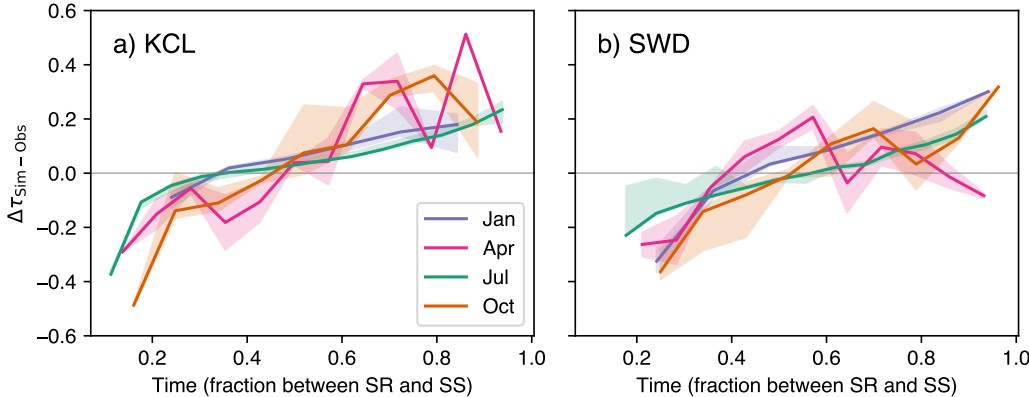

**Figure 7.** Bulk transmissivity difference (model-observation) of 60 min values median (line) and interquartile range (shading) during the daylight hours (normalised between sunrise (SR: 0) and sunset (SS: 1)) during four periods of the year (Table 4) at: (a) KCL and (b) SWD. See Sect. 2.5.

cause underestimation of the peak $K_\downarrow$ values in summer (Fig. 8c cf. Fig. 8g ), as the WRF overestimated transmissivity is larger in the afternoon (Fig. 7). Thus using a single bulk correction will overcorrect $K_\downarrow$ in other periods, notably near midday.

On cloudy days, the correction also improves the $K_\downarrow$ performance. The MAE and MBE, as well as the HR are enhanced, despite the correction parameter not being derived for cloudy conditions (Fig. 9). The MBE is reduced by more than $50\%$ for all simulation periods at both sites (except October 2012 at SWD) after correction. In general, we deem such correction necessary and effective for land surface modules in the WRF system (v4.0); hence we use $K_\downarrow$-corrected simulation results throughout the following analyses.

### 3.4.2 Online and offline simulated surface energy balance fluxes

Although our focus is on the online WRF-SUEWS system it is useful to compare its performance to the offline (i.e. standalone SUEWS) version. As we force the latter directly with observations (refer to the "Atmospheric forcing" variables in Fig. 1), removing potential forcing errors from the online system, we expect better performance and hence use this as a benchmark. The same simulation periods were used for the online and offline evaluation (Table 4). Given the SUEWS v2018c kernel is almost the same as the v2016 kernel, the offline performance is very similar to the values reported by W16. The largest difference may be associated with an update to the OHM calculations, as v2016 uses the two preceding hours mean net all-wave radiation values, while v2018c uses the step-size weighted average of two successive timesteps.

Overall, the offline (upper row, Fig. 12) performance is better than online (lower row) at both sites for all fluxes. Although offline $K_\downarrow$ should have no error, slight deterioration in performance occurs (Ward et al., 2016), because the high-resolution (e.g., $1\,\text{min}, 5\,\text{min}$) forcing is not used but rather 60-min means are interpolated to $5\,\text{min}$ and subsequently re-averaged to 60 min for evaluation. The offline $K_\downarrow$ HR is greater than 0.93 for all four seasons at both sites, whereas the online $K_\downarrow$ HR are

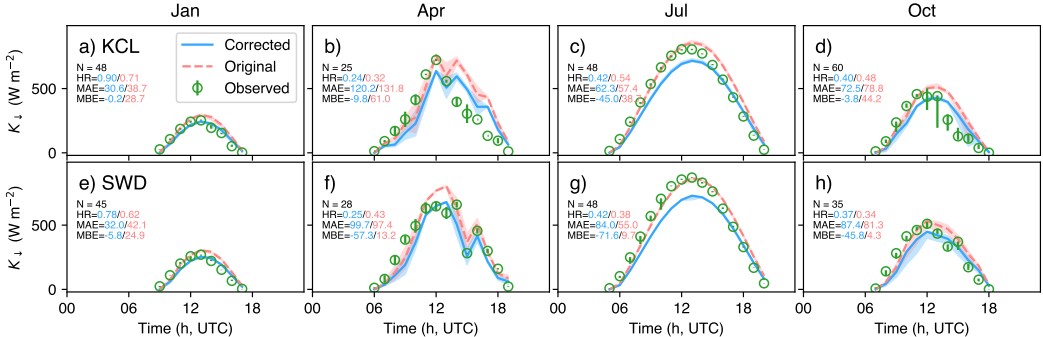

**Figure 8.** Daytime 60 min shortwave incoming radiation ($K_\downarrow$) fluxes during clear sky days. Observed data is marked with median values (circles) and interquartile range (IQR, vertical lines), while simulations by WRF-SUEWS are illustrated with median (lines) and interquartile range (shading). Both the original (solid) and corrected bulk transmissivity (dashed) are utilised across four two-week periods (refer to Table 4) at the (a-d) KCL and (e-h) SWD sites. Scarce clear sky periods occurred in April. For additional details on metrics and units, refer to Sect. 3.3.

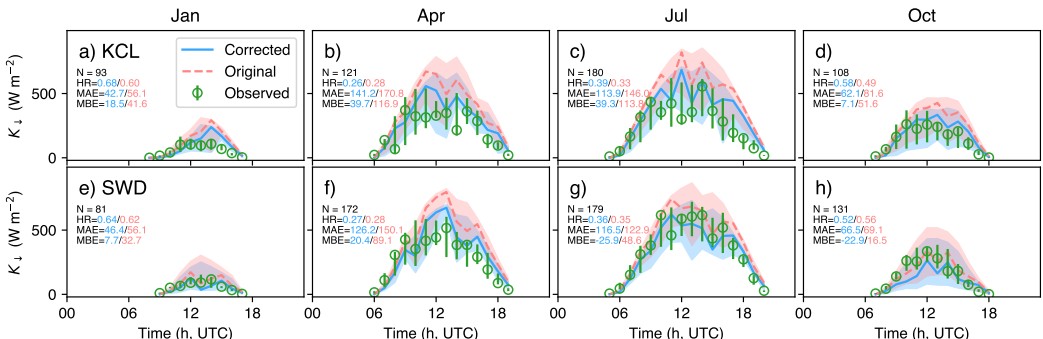

**Figure 9.** Same as Fig. 8, but for cloudy days.

0.25 to 0.75 (Fig. B1-B2) with HR $< 0.5$ and MAE $> 60$ W m$^{-2}$ in April and July (Fig. 12). The model performance in $K_\uparrow$

is comparable at both sites (Fig. 12); the offline mode proves to be superior to the online mode - this is not surprising as it corresponds to the model's performance in $K_\downarrow$.

     The outgoing longwave radiation $L_\uparrow$ is modelled well (MAE $< 12.6$ W m$^{-2}$) both offline and online. But like $K_\downarrow$, $L_\downarrow$ is poorer for both cases (MAE online: 28.6; offline: 28.2 W m$^{-2}$) and its diurnal patterns is quite captured by the model correctly - the offline mode lacks sufficient diurnal variability while the online mode shows general underestimation (cf. Figs. 10 and 11).

Net all-wave radiation ($Q^*$) is impacted mostly by $K_\downarrow$ during the day making the WRF $K_\downarrow$ correction important. The intra-annual range of MAE for the online $Q^*$ simulations is smaller for KCL $\left(21 - 64\text{ W m}^{-2}\right)$ than at SWD $\left(37 - 72\text{ W m}^{-2}\right)$, so the turbulent fluxes at SWD start with a potentially greater error.

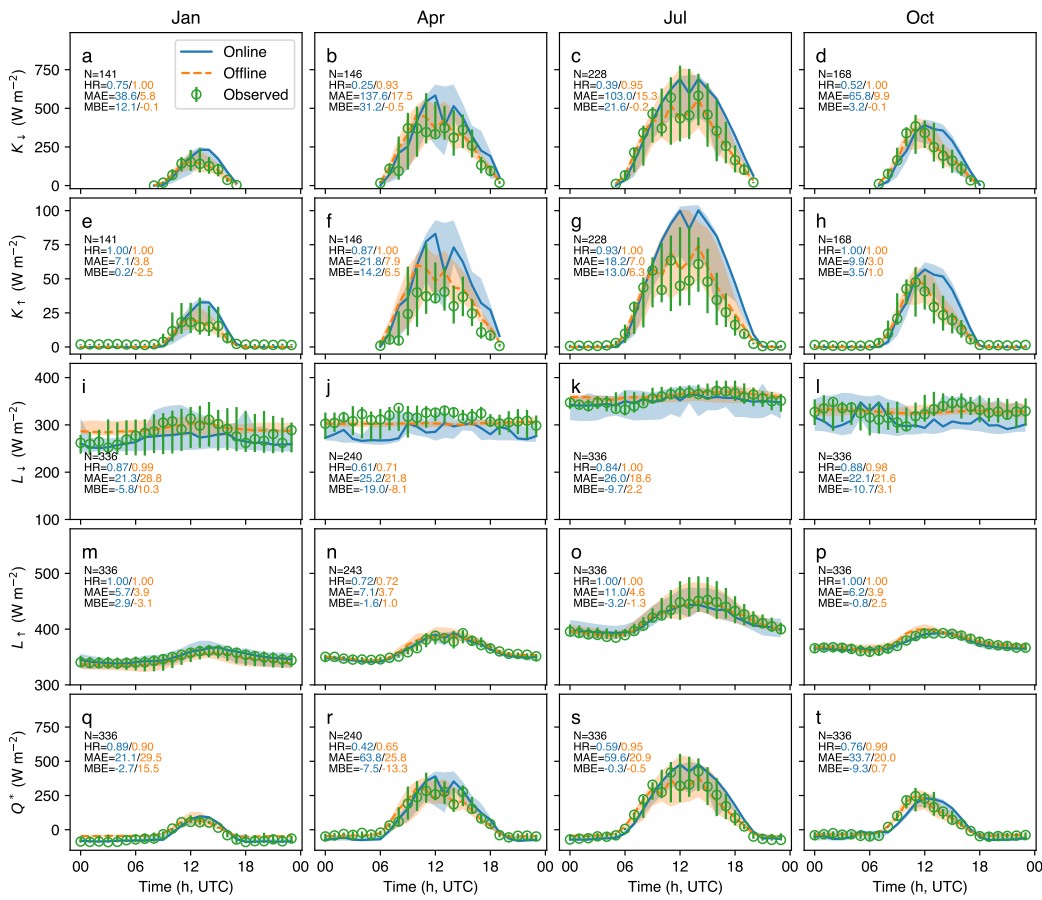

**Figure 10.** Diurnal pattern of simulated (median: online - line, offline - dashed; IQR: shading) and observed (median: circle, IQR: vertical line) incoming (a-d) shortwave ($K_\downarrow$) and (e-h) longwave ($L_\downarrow$), outgoing (i-l) short ($K_\uparrow$,) and (m-p) longwave ($L_\uparrow$), and (q-t) net all-wave ($Q^*$) for four two-week periods in different seasons (Table 4) at KCL. "N" indicates number of hourly data points used in the analysis; other metrics are defined in Sect. 3.3.

Clear seasonality in MBE for the turbulent heat fluxes ($Q_E$ and $Q_H$) occurs at both sites (Fig. 12). At KCL there is a positive bias of varying magnitude (6 to 47 W m$^{-2}$) for both fluxes, whereas at SWD $Q_E$ is generally underestimated ($-6$ to

$-4$ W m$^{-2}$) and $Q_H$ is overestimated (underestimated) by 25 W m$^{-2}$ ($-31$ W m$^{-2}$) in January (July).

Overall, WRF-SUEWS better predicts $Q_E$ than $Q_H$ at both sites (MAE$_{Q_E}$ vs. MAE$_{Q_H}$ : 26 W m$^{-2}$ vs. 45 W m$^{-2}$ at KCL; 18 W m$^{-2}$ vs. 62 W m$^{-2}$ at SWD). The diurnal performance for the turbulent heat fluxes and Bowen ratio ($\beta = Q_H/Q_E$) is similar for both offline and online runs for both KCL and SWD (Figs. 13 and 14). The $\beta$ indicates if the turbulent heat fluxes are correctly partitioned (i.e., making radiation performance less critical). The WRF-SUEWS $\beta$ agrees well with the observations

at both KCL and SWD. However, when both fluxes are small $\left(< 10 \text{ W m}^{-2}\right)$ there are both larger observational errors and ratios change rapidly. Under these conditions in January, $\beta$ is overestimated (nocturnal at KCL; all times at SWD).

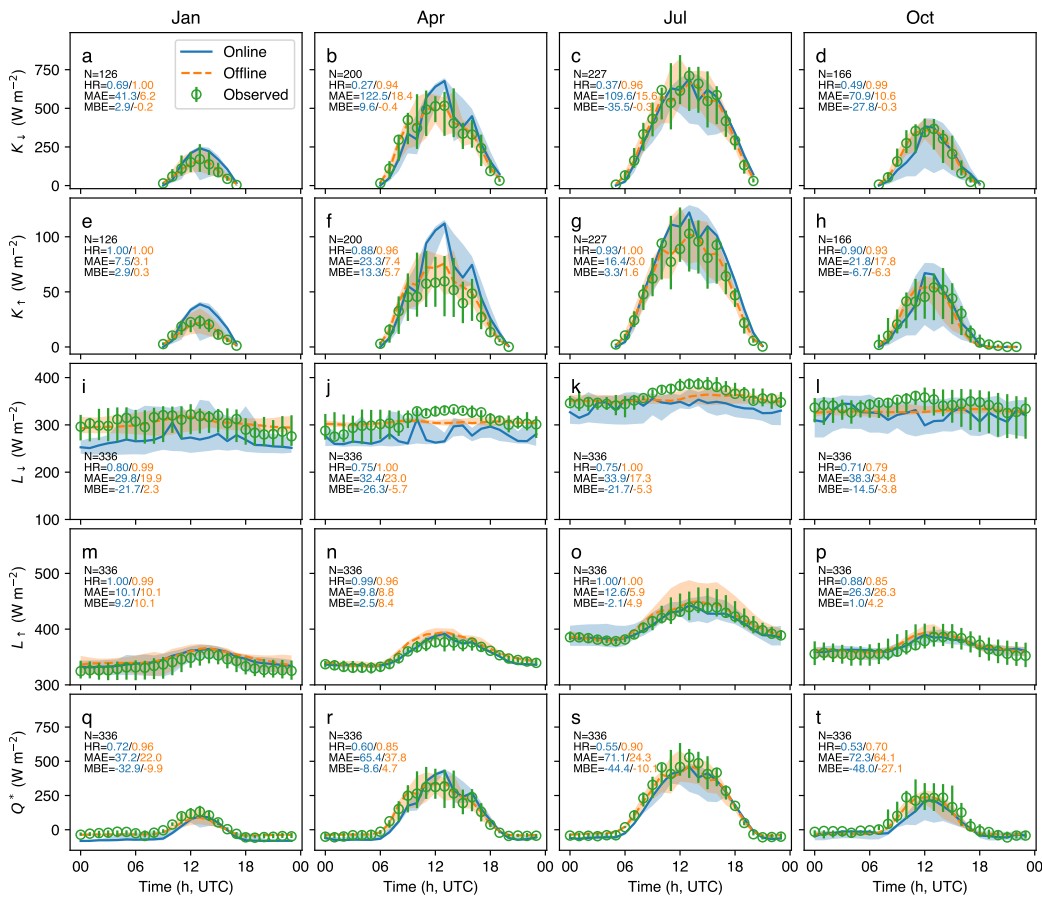

**Figure 11.** As 10, but for the SWD site.

To set these results in context, it is useful to compare them to the results of other urban land surface models that have been evaluated in this region (e.g. SLUCM: Loridan et al., 2013; Tsiringakis et al., 2019; Best-1T and MORUSES with JULES: Hertwig et al., 2020). Loridan et al. (2013, hereinafter L13) focussed on the 3 June 2010 using KCL observations to evaluate

the SLUCM (Kusaka et al., 2001) in WRF (Chen et al., 2011). Hertwig et al. (2020, H20) evaluated two urban schemes with JULES (Best et al., 2011), the Best-1T (Best et al., 2006) and MORUSES (Porson et al., 2010), over the 2011-2013 period at both the KCL and SWD sites. The evaluation strategies differ: L13 considers both online and offline performance but with different surface information; H20 is offline only, but considers a range of configurations. For comparison we consider their more "advanced" configurations (L13: online, UZE - Urban Zones for Energy partitioning - for plan area index-based surface

categorisation; H20: offline, a baseline configuration CTRL-B and a more sophisticated one CTRL-M with the more realistic anthropogenic forcing and detailed land cover information) using appropriate periods.



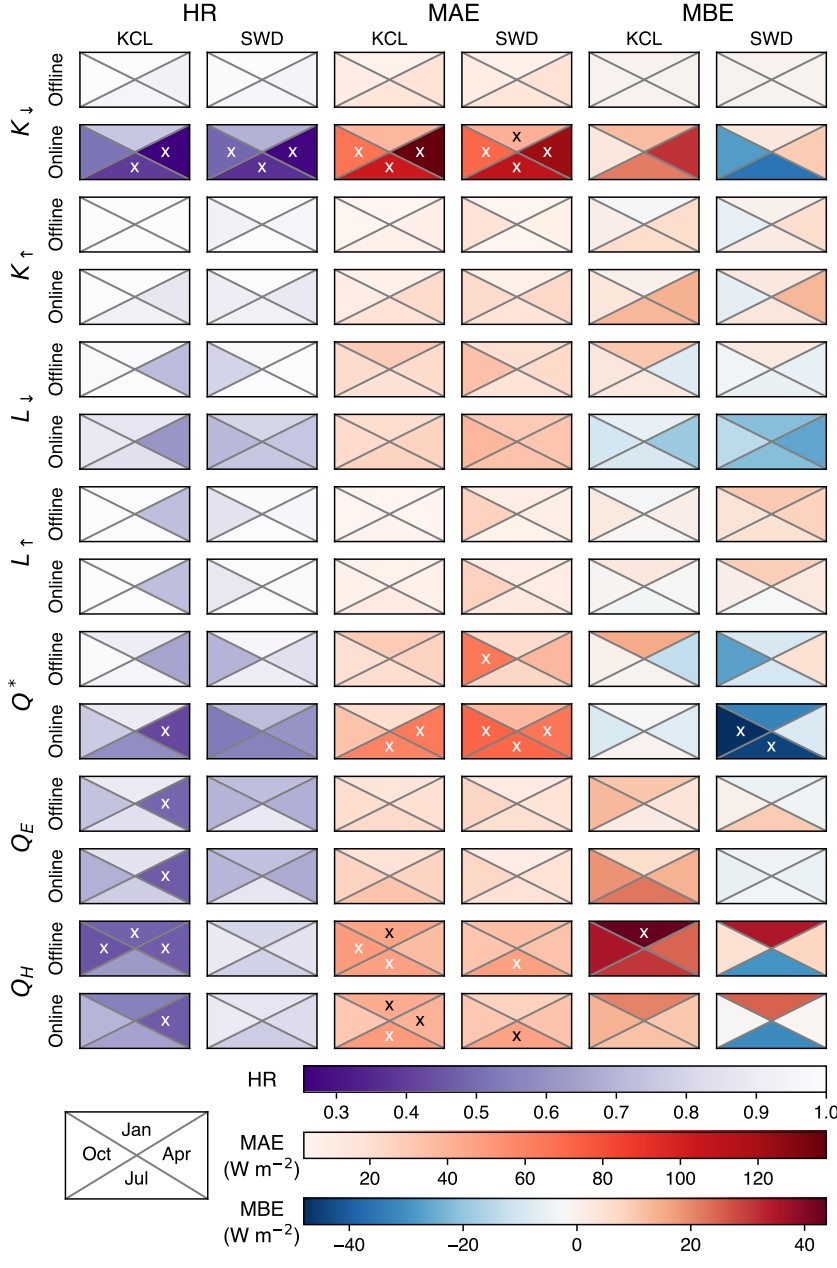

**Figure 12.** Model performance for (rows) radiative ($K_\downarrow, K_\uparrow, L_\downarrow, L_\uparrow$ and $Q^*$) and turbulent heat ($Q_E$ and $Q_H$) fluxes for (rows) simulated in online and offline modes at (columns) KCL and SWD assessed using (columns) three metrics (HR, MAE, MBE: colour - darker poorer performance) for four periods (triangles). Triangles marked ( x ) indicate the model performance can be categorised unsatisfactory based on the one of following criteria: $HR < 0.5, MAE > 40\ W\ m^{-2}$, and $|MBE| > 40\ W\ m^{-2}$. Metrics are defined in Sect. 3.3.



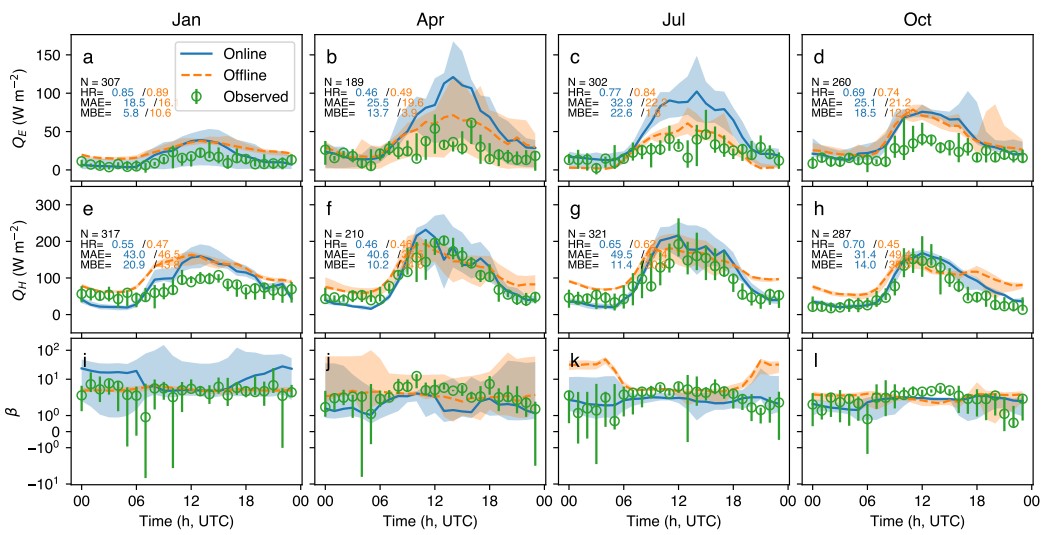

**Figure 13.** Diurnal pattern of simulated (median: online - line, offline - dashed; IQR: shading) and observed (median: circle, IQR: vertical line) fluxes: (a-d) sensible ($Q_H$) and (e-h) latent ($Q_E$) heat fluxes, and (il) Bowen ratio ($\beta = Q_H/Q_E$) (hourly median) for (columns) four two-week periods in different seasons (Table 4) at KCL. Metrics defined in Sect. 3.3

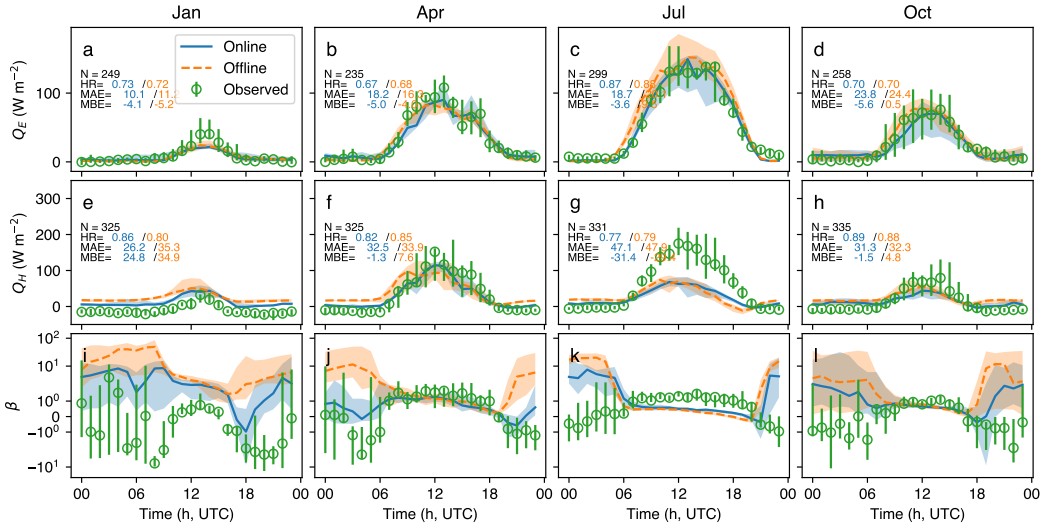

**Figure 14.** Same as Fig. 13, but for SWD.

WRF-SLUCM performance at KCL for $Q_E$ is better (WRF-SUEWS MAE $= 33$ W m$^{-2}$); noted that the L13 study covers only 1 day (cf. our summer 14 days). Similar to SUEWS, SLUCM's online mode performs slightly worse than its offline mode with respect to RMSE (online vs. offline): 82.8 W m$^{-2}$ vs. 82.6 W m$^{-2}$ for $Q_H$; while for $Q_E$, 16.4 W m$^{-2}$ vs.



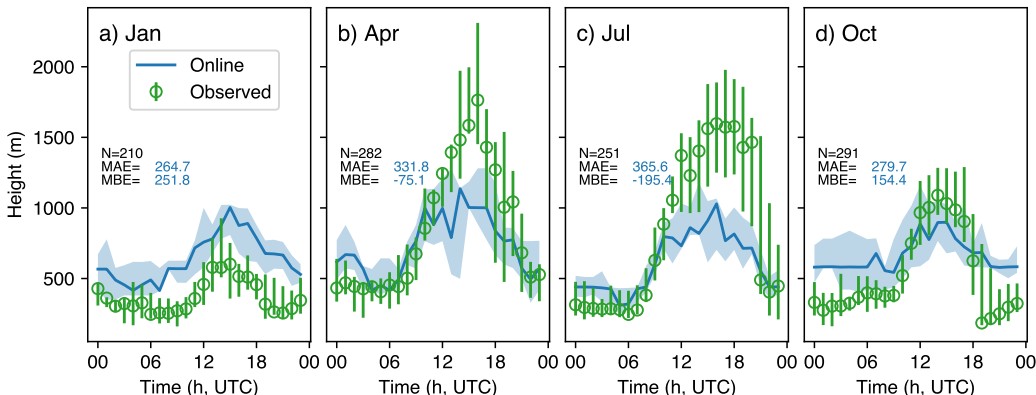

**Figure 15.** As Fig. 13, but for MLH (observed) and MH (online, Sect. 3.3) at MR in London.

$14.2 \text{ W m}^{-2}$. At KCL, the annual offline $\text{MAE}_{\text{SUEWS}}$ for $Q_E$ is similar to CTRL-B $\left(\sim 20 \text{ W m}^{-2}\right)$ but larger than CTRL-M $\left(\sim 15 \text{ W m}^{-2}\right)$, whereas all three model configurations have similar performance at SWD $\left(\sim 15 \text{ W m}^{-2}\right)$. For $Q_H$, all three models are similar at KCL $\left(\text{MAE} =\sim 40 \text{ W m}^{-2}\right)$ but at SWD, $\text{MAE}_{\text{SUEWS}}$ is larger $\left(\sim 30 \text{ W m}^{-2}\right)$ than for the two H20 models $\left(\sim 20 \text{ W m}^{-2}\right)$.

Although not directly comparable, the SUEWS performance appears to be consistent with both offline and online perfor-
mance of other urban land surface models in this area. Attribution of bias differences (e.g. different parameterisations, config-
urations, land cover information) is out of the scope of this study but is the focus of model comparison studies (e.g. Grimmond
et al., 2010a, b).

### 3.4.3 Boundary layer depth

To assess the boundary layer depth the modelled MH (Sect. 3.3) is compared to the observed MLH at MR in London (Fig. 15).
Generally, WRF-SUEWS underestimates daytime MH in all seasons except for winter (Fig. 15a) with $\text{MAE} > 300 \text{ m}$ in the
warmer periods (Fig. 15b,c). WRF-SUEWS slightly overestimates MH at night. The MBE varies between -195 and 252 m
for the four periods. These values are smaller than WRF-SLUCM (with multiple PBL schemes, -288 to 539 m) simulations
over Greater Paris evaluated using radiosonde observations (Kim et al., 2013). Similarly, evaluating WRF using eight different
PBL schemes in Barcelona, Banks et al. (2015) found daytime MH to be underestimated (cf. elastic backscatter lidar), with the
largest relative bias of -48%. This is comparable to our summertime results (Fig. 15c).





## 4    Application of WRF-SUEWS: impacts of anthropogenic heat (QF) on atmospheric boundary layer

### 4.1    Modelled variability of anthropogenic heat emissions

The WRF-SUEWS model demonstrates satisfactory performance across various seasons and for two distinct urban areas
(Sect. 3.1). Therefore, we employ it to examine the influence of $Q_F$ on the atmospheric boundary layer - a unique charac-
teristic of the coupled WRF-SUEWS system compared to the standalone SUEWS - in d03 (Fig. 5) during April 2012, before
the Olympics disrupted typical patterns in July. As daylight saving has begun by this time of the year, people's activities are
shifted an hour earlier (e.g. typical workday is 08:00-16:00 UTC, or 09:00-17:00 BST local time). Peak $Q_F$ emissions occur
in central London (Fig. 16) during the daytime ($> 300$ W m$^{-2}$, Fig. 16c). By the late afternoon and through the night (20:00-
05:00 UTC) the values in these areas (Fig. 16b) are smaller ($< 130$ W m$^{-2}$, Fig. 16d). The areas where large values occur
differ between night and day.

At night the peaks occur across a larger area beyond central London, with some larger values towards the outskirts (Fig. 16b
cf. Fig. 16a). The nocturnal $Q_F$ mean in d03 is smaller $\left(17.1 \text{ W m}^{-2}\right)$ than the daytime $\left(19.8 \text{ W m}^{-2}\right)$ but more spatially
consistent (Fig. 16d cf. Fig. 16c).

These spatiotemporal patterns (Fig. 16a, b) are largely consistent with previous studies in London. Daily peak values differ
between studies because of model grid cell size (Lindberg et al., 2013), and efforts over time to reduce carbon emissions and
therefore energy use (Lindberg et al., 2013; Ward and Grimmond, 2017). Peak values (grid size) vary between 120 W m$^{-2}$
(3.2 km$^2$, Capel-Timms et al., 2020), $\sim 150$ W m$^{-2}$ (1 km$^2$, Hamilton et al., 2009; 1 km$^2$, Bohnenstengel et al., 2013) and
210 W m$^{-2}$ (3.2 km$^2$, Iamarino et al., 2011).

### 4.2    Feedbacks from anthropogenic heat emissions

To assess the feedback, we focus on two example grids with contrasting population density profiles:

- a central business district (CBD) area, within the City of Westminster (grid C, Fig. 16), with tall buildings (roughness
  length $z_0 = 2.0$ m, zero plane displacement $z_d = 13.9$ m; calculated using the rule-of-thumb approach as in Grimmond
  and Oke, 1999), low fraction of vegetation ($f_{\text{VEG}} = 13.1\%$), and large daytime population density $\left(763 \text{ inh ha}^{-1}\right)$ but
  small nocturnal density $\left(111 \text{ inh ha}^{-1}\right)$.

- an inner-city residential area, within the Borough of Islington (grid R, Fig. 16), with $z_0 = 0.8$ m, $z_d = 5.6$ m, $f_{\text{VEG}} = $
  43.3%, and nocturnal population density $\left(170 \text{ inh ha}^{-1}\right)$ slightly greater than the daytime density $\left(150 \text{ inh ha}^{-1}\right)$.

Workday time series reveal differences in $Q_F$ timing and magnitude between the two grids (Fig. 17a). In grid C, $Q_F$ is
significantly higher in general, staying at a rather consistent level between the morning and evening peaks, whereas the flus
is lower in grid R and shows a reduction in emission between the two peaks. However, the second peak in R is much later
(R: $\sim 21:00$ UTC, C: $\sim 17:00$ UTC, Fig. 17a, red). Both grids have similar $Q_E$ (Fig. 17a) despite the 30% difference in
vegetation fraction, attributable to the low LAI in April (Kotthaus and Grimmond, 2014a, b; Ward et al., 2013). The net
radiation ($Q^*$, Fig. 17a) values are also similar. The larger $Q_F$ in grid C enhances $Q_H$ by $\sim 220$ W m$^{-2}$ over grid R during the



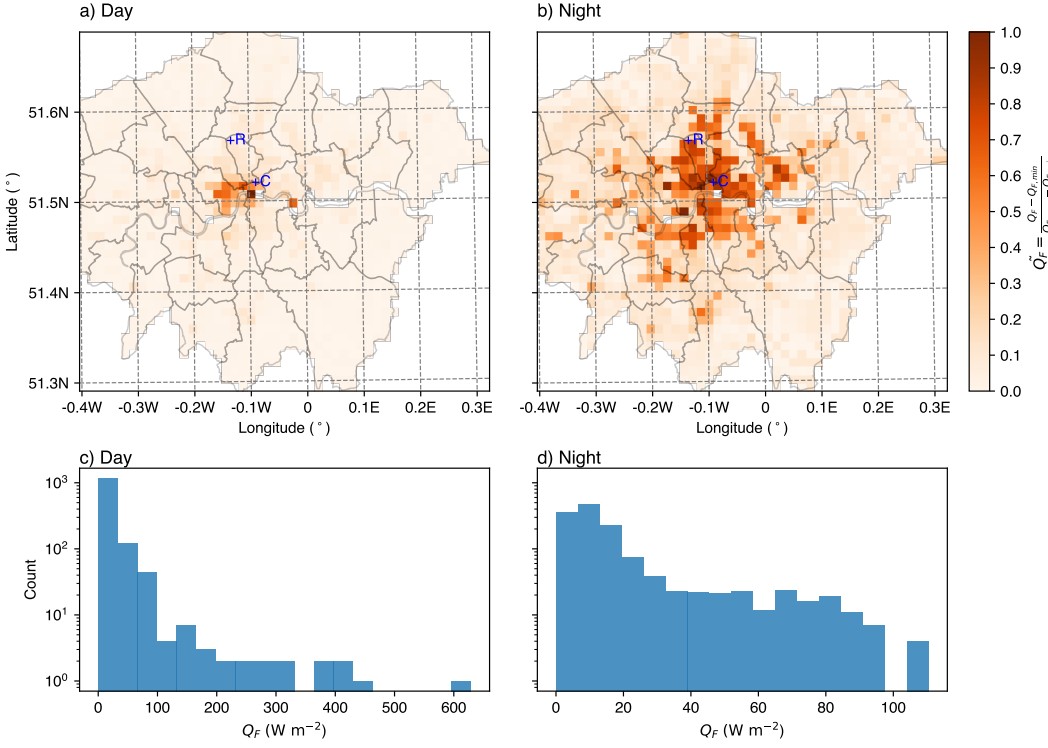

**Figure 16.** April (Table 4) weekday normalised anthropogenic heat flux in London (grey lines are boroughs) during (a) main working hours in the day (08:00 - 16:00 UTC), (b) night (20:00 - 05:00 UTC), and (cd) the respective distributions of their actual values (in W m$^{-2}$). Grids analysed in more detail are indicated (blue): C - commercial/business, R - residential.

middle of the day on average, which contributes to an increase in midday MH of 100–200 m compared to grid R (700–900 m above ground level; Fig. 17b, c).

The contrasting $Q_F$-induced heating alters several atmospheric variables within the urban boundary layer: daytime near surface air ($< 100$ m agl) in grid C is warmer (positive difference, $\sim 0.3$ K, Fig. 17b) and drier (negative difference, $\sim 0.2$ g kg$^{-1}$, Fig. 17c) than in grid R. At night, the air temperature in grid C stays warmer at lower altitudes but with cooler and drier air aloft with PBL (Fig. 17). These results are consistent with other WRF-SLUCM-based studies stating that urban heating leads to warmer and drier near-surface atmosphere (e.g. Zhang and Chen, 2014; Zhang et al., 2015). The $Q_F$ impacts

are of similar magnitude to those linked to urban greening (but with an inverse effect): for instance, with all roofs vegetated in Beijing, the near-surface atmosphere is cooled by $\sim 1$ K but moistened by $\sim 0.8$ g kg$^{-1}$ during a heat wave period even when anthropogenic heat is accounted for (Sun et al., 2016); this suggests that urban greening may help mitigate some effects of anthropogenic heat emissions.





**Figure 17.** April (Table 4) weekday diurnal pattern in two grids (C and R, Fig. 16) of (a) mean surface energy fluxes, and (b, c) median MH and median difference (colour, C minus R) with height of (b) potential temperature $\theta$ and (c) specific humidity $q$.

## 5 Concluding Remarks

Through coupling the SUEWS urban land surface model to WRF, urban-atmosphere interactions can be explored more fully explored than when using the standalone SUEWS. The new Fortran subroutine SuMin interfaces SUEWS with the



land surface driver of WRF. The WSPS pre-processor incorporates SUEWS specific parameters into the `wrfinput` and `namelist.suews` files for WRF-SUEWS simulations. The coupling is designed to permit sustainable development of SUEWS so that regular enhancements of SUEWS can be seamlessly incorporated into the coupled system.

Evaluation of the coupled WRF-SUEWS system is performed at two UK sites: dense central London (KCL) and suburban/residential site in Swindon (SWD) across four seasons. It generally shows a good capacity of the coupled system to simulate the surface energy balance fluxes and mixing height in all periods. The performance compares well to other WRF studies in urban settings (Banks et al., 2015; Kim et al., 2013; Loridan et al., 2013). Better performance is found (i) at the suburban site SWD than the densely built-up KCL for the turbulent heat fluxes; (ii) during clear sky conditions for radiative

than turbulent heat fluxes; (iii) using a bulk atmospheric transmissivity for incoming shortwave radiation (than without).

WRF-SUEWS' capability allows analyses of spatial and temporal variations over heterogeneous urban areas compared to existing WRF urban schemes (e.g. SLUCM, BEP, etc.) that can only resolve a limited number of urban classes. Critically, the SUEWS capability allows for dynamic feedbacks from human activities (e.g. heat, water, phenology, snow related). The influence of anthropogenic heat $Q_F$ on the boundary layer in April prior to leaf growth in Greater London influences the

sensible (more than the latent) heat fluxes. The larger $Q_F$ and $Q_H$ in central London are associated with warmer and drier air and deeper mixing heights during the day but not at night. The WRF-SUEWS evaluation needs to be expanded to other urban settings. Results suggest that the system has a great potential to help advance our understanding of the role of urban surface heterogeneity. As SUEWS is already integrated with many other models (e.g. building energy parameterisations and thermal comfort simulations, Table 1), WRF-SUEWS can help decision makers involved in a wide range of integrated urban services

to identify the spatiotemporal distribution of near-surface meteorology (e.g. heat-induced risks) across a city.

*Code and data availability.* The snapshots of input data and source code for WRF-SUEWS used in this paper have been archived on Zenodo at https://doi.org/10.5281/zenodo.7957903 (Sun et al., 2023a) and https://zenodo.org/record/8137708 (Sun et al., 2023b), respectively. The up-to-date version of WRF-SUEWS is available at https://github.com/Urban-Meteorology-Reading/WRF-SUEWS (last access: 16 May 2023).

*Author contributions.* TS led the development of WRF-SUEWS with significant contributions from HO and ZL. TS and HO performed the evaluation. TS, HO and SG drafted the manuscript and all authors contributed to review and editing of the manuscript.

*Competing interests.* The authors declare that they have no conflict of interest.

*Acknowledgements.* This work has been supported by the Newton Fund/Met Office CSSP-China: HighResCity AJYG-DX4P1V), the Natural Environment Research Council (Independent Research Fellowship, grant no. NE/P018637/1 and NE/P018637/2; COSMA, grant no.



NE/S005889/1; ClearfLo, NE/H003231/1; Studentship: NE/H52479X/1), the European Research Council (Synergy: urbisphere 855005, FP7: BRIDGE 211345) and the National Natural Science Foundation of China (grant no. 41975006). HCW is supported by the Austrian Science Fund (grant no. M2244-N32 and V888-N).





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
