# Peer review of "WRF (v4.0)-SUEWS (v2018c) Coupled System: Development, Evaluation and Application"

_Geoscientific Model Development, 2023_

## Author Comment (AC1)

**Response to reviews on manuscript GMD-2023-117**

We appreciate the insightful comments from reviewers that have remarkably improved the quality of our manuscript. Please find below:

- our point-to-point responses (Sans Serif font in blue) to reviewer comments (RCs); and
- excerpts of revisions in salmon with a grey background , where necessary.

**Response to the reviewers**

**Reviewer 1**

The manuscript presents the coupling Surface Urban Energy and Water Scheme (SUEWS) into the Weather Research and Forecasting (WRF) model. The coupled WRF-SUEWS system shows good performance in simulating fluxes and mixed layer height compared to observations in two UK sites as well as other urban-focused WRF research. The integrated system has also been employed to investigate the impacts of anthropogenic heat flux emissions on the boundary layer dynamics within Greater London. The research is systematically structured and well presented.

Reply: We appreciate your recognition of our work.

**RC 1.1** Lines 53-54: Please elucidate the distinction between SUEWS and the land surface mentioned. Does SUEWS replace the entire land surface module in WRF?

**Reply**: Yes, SUEWS would replace the entire land surface module in WRF as it was introduced as a standalone option for the land surface part in WRF.

We have clarified this in Sect. 2.1 as follows:

The coupling between WRF and SUEWS occurs via the biophysical interactions between the **land surface** – with SUEWS introduced as new land surface module option – and other physics modules in WRF:

**RC 1.2** Lines 217-219: The notation WREE seems ambiguous. Is it intended to be WRF? Also, ensure that each term in Eq. 20 is clearly defined, especially if not introduced earlier.

Reply: Thank you for pointing out the typos, which have been corrected in the revision as follows:

... between  $\tau_{\rm WRF}$  and  $\tau_{\rm obs}$  ...

Also, terms used in Eq. 20 has been clarified in its introductory sentence as follows:

As  $\tau$  can vary seasonally (e.g. the cases in London and Swindon as shown in Table 4) we determine the median clear sky difference  $(\Delta \tau_c)$  between  $\tau_{\text{WRF}}$  and  $\tau_{\text{obs}}$  from analysis of clear sky days observations around the peak  $K_{\downarrow}$  (which occurs between 40% and 60% of the daylight hours). The  $K_{\downarrow}$ forcing (Fig. 1) for SUEWS in WRF ( $K_{\downarrow,W-S}$ ) is then corrected using the original one produced by WRF ( $K_{\downarrow,WRF}$ ) as: ...

**RC 1.3** Line 245: Table 2 seems to lack blue dots. Are you referring to Figure 6?

**Reply**: The blue dots indeed refer to those in Fig. 6. After scrutiny, this was found to be related to a LATEX compilation issue and has been fixed in the revised manuscript.

**RC1.4** Figure 6: Is IMP indicative of impervious surfaces? Kindly specify.

Reply: The meaning of "IMP" is now added in the caption of Fig. 6 as follows:

The individual grid cells are categorised first by the greatest land cover fraction with impervious (IMP) split between paved surfaces (PAV) and buildings (BDG).

**RC 1.5** Lines 275-276: Which observational forcings are utilized for the SUEWS spin-up? Please clarify.

**Reply**: The observational forcing variables for the spin-up include: incoming solar radiation  $K_{\downarrow}$ , incoming long-wave radiation  $L_{\downarrow}$ , precipitation P, air temperature  $T_a$ , atmospheric pressure  $p_a$ , relative humidity RH, and wind speed U. They align with those used in the driving typical SUEWS runs – both online and offline modes – and other land surface modules in WRF as indicated by the purple boxes in Fig. 1 in the manuscript.

This has been clarified in Sect. 3.2 as follows:

(refer to the purple boxes in Fig. 1 and related notations in Sect. 2.2 for details about the atmospheric forcing variables as well as Table 2 for surface property settings)

**RC 1.6** Lines 296-298: Could you elaborate on the rationale behind selecting aerosol-derived MLH observation and WRF MH as evaluative metrics for the boundary layer depth?

**Reply:**

We chose the aerosol-derived mixed layer height (MLH) as an evaluative metric for the height of atmospheric boundary layer (ABL) mainly due to two reasons:

- Data Availability: The aerosol-derived mixed layer height (MLH) dataset is the best observational data we have in London to represent the height of ABL.
- Comparable nature: While we acknowledge the intrinsic differences between MLH and mixing height (MH) concerning the physical processes they represent – MH captures the well-mixed characteristics of the ABL, whereas MLH reflects its dynamic mixing features – both MLH and MH can serve as indicators of the height of ABL.

Clarified in Sect. 3.3 as follows:

The mixed layer height (MLH), derived from continuous high resolution (15 s and 10 m) attenuated backscatter observed with a Vaisala CL31 ceilometer at Marylebone Road (MR) in London (Kotthaus and Grimmond, 2018a,b; Kotthaus et al., 2016), is used to evaluate the model's ability to predict atmospheric boundary layer (ABL) dynamics. The MLH values have been compared to AMDAR, or Aircraft Meteorological Data Relay (the median difference between inversion heights and MLH is 346 m based on all time periods; more evaluation results refer to Kotthaus and Grimmond (2018a)). ... While the comparison of the aerosol-derived MLH from observations and the turbulence-based mixing height diagnosed from the model output (WRF PBL, hereafter referred to as WRF MH) may be affected to systematic differences (e.g. those associated with vertical resolution as suggested by Kotthaus et al. (2023)), the comparable nature of MLH and MH enables the former to serve as a proxy for examining the latter modelled by WRF.

**RC 1.7** Figure 7: The bulk transmissivity difference seems to rise progressively from sunrise to sunset. Could you elucidate the underlying reason?

**Reply**: This is attributed to using a bulk correction of transmissivity based on the midday observation throughout a day, hence morning and afternoon bias occur; creating the progressively ascending trend in  $\Delta \tau$  as seen in Fig. 7.

Clarified in Sect. 3.4.1 as follows:

Thus using a single bulk correction will have a diurnal bias in  $K_{\downarrow}$  from under(over)-correction in the morning (afternoon). This is evident in the ascending trend of  $\Delta \tau_{\text{Sim-Obs}}$  (cf. Fig. 7).

**RC 1.8** Lines 345-346: The authors mentioned that the offline mode outperforms the online mode due to the model's performance in incoming shortwave radiation. Given the study's primary objective of evaluating the coupled system (i.e., the online mode), what prompts the emphasis on reducing offline shortwave radiation errors? Would such corrections augment the online mode's efficacy?

**Reply**: We are *NOT* correcting the radiation in the *offline* mode, rather the correction (Sect. 2.5) is in the *online* system to reduce the bias in the overestimated  $K_{\downarrow}$  (Jimenez et al., 2016; Lapo et al., 2017) and it cascading through the other fluxes.

As for the efficacy of this online system, if confined to the land surface module, it would correct the overestimated  $K_{\downarrow}$ , thereby improving predictions of downstream energy fluxes (e.g.,  $Q_E$ ,  $Q_H$ , etc.). However, given this inherent complexity, any specific improvement may not necessarily enhance performance or impact other aspects (e.g. precipitation prediction).

**RC 1.9** Lines 358-359: Why does the accurate partitioning of turbulent heat fluxes makes radiation performance less critical?

**Reply:**

As discussed in Sect. 2.5, as WRF generally overestimates incoming solar radiation and this cascades through all surface energy fluxes (see RC1.8). Our wording "making radiation performance less critical

" is poor – our intent is to reduce this bias in the WRF-SUEWS system. Radiation biases affect the absolute turbulent heat fluxes, but their relative partitioning much less so.

Revised in Sect. 3.4.2 as follows:

The  $\beta$  indicates the turbulent heat fluxes are correctly partitioned, suggesting the model's robustness in turbulent heat flux partitioning even when there are variations in radiation accuracy (i.e., making the skill in simulating the absolute radiation fluxes less critical).

**RC 1.10** Figure 14: The model seems to inadequately represent the Bowen ratio compared to observations, especially during nights in July and October in SWD. Could you shed light on this inconsistency?

**Reply**: Uncertainty in eddy-covariance (EC) flux measurements is greater at night because of the lower turbulence reducing the fluxes (Järvi et al., 2018; Mahrt et al., 2012). As  $\beta$  is a ratio of small numbers a small absolute variation has a large impact on the ratios.

We add some references in Sect. 3.4.2 as follows:

The WRF-SUEWS daytime  $\beta$  agrees well with the observations at both KCL and SWD. However, when both fluxes are small (< 10 W m-2) there are both larger observational errors (e.g. uncertainties due to nocturnal weak turbulence; Järvi et al., 2018; Mahrt et al., 2012) and ratios change rapidly. Under these conditions, nocturnal  $\beta$  is overestimated (January at KCL; all seasons at SWD).

**RC 1.11** Line 391: Change QF to QF?

**Reply**: Corrected as suggested.

**RC 1.12** Line 418: Rectify flus to flux.

**Reply**: Corrected as suggested.

**RC 1.13** Lines 420-422: The 30% discrepancy in vegetation fraction—does it signify an annual or monthly average? If it's an annual average, could you specify the discrepancy for April? Furthermore, could you expound on the decision to spotlight April when investigating the effects of anthropogenic heat on the atmospheric boundary layer? Are other summer or winter months considered?

**Reply**: The 30% difference in vegetation fraction is invariant throughout a year as configured in the simulations; although LAI does vary (but does not impact  $Q_F$ ).

Regarding the month selected in the analysis: We avoid July 2012 as the Olympics modified many human activities. Winter months have large  $Q_F$  in the UK, but April still can be cold but warm day can have very large sensible heat fluxes (Kotthaus and Grimmond, 2014). Given this is a model development paper, we select this month to explore the  $Q_F$  impacts. With the new system more cities and periods can be explored in more depth.

We have added this in Sect. 5 as follows:

The WRF-SUEWS evaluation should be expanded to other urban settings, timeframes and synoptic conditions, and further applications explored (e.g.  $Q_F$  impacts on urban-atmospheric interactions).

**RC 1.14** Figure 16: Please provide more details regarding the computation of anthropogenic heat (QF), particularly the exact meaning of Qmax and Qmin. Furthermore, the use of normalized anthropogenic heat in Figures (a) and (b) is puzzling. This format complicates direct comparisons of anthropogenic heat during daytime and nighttime.

**Reply**: Please refer to Eq. (3) and related descriptions in Sect 2.2 for details of  $Q_F$  calculation.  $Q_{F,\max/\min}$  indicates the the maximum/minimum  $Q_F$  values across the study area (i.e. the Greater London Area as shown in Fig. 16a and 16b) at respective times, which are used to normalise the  $Q_F$  for manifesting the spatial contrasts.

To retain these, but for clearer interpretation, we have added the maximum/minimum values in the histogram subplot and modified the caption of Fig. 16 as follows:

**Figure 16**. April (Table 4) weekday anthropogenic heat flux normalised by difference between maximum and minimum  $Q_F$  (i.e.  $Q_{F,\max} - Q_{F,\min}$ ) in London (grey lines are boroughs) during different periods: (a) main working hours in the day (08:00 - 16:00 UTC) and (b) night (20:00 - 05:00 UTC) with (c, d) the respective distributions of their actual values (in W m-2). Grids analysed in more detail are indicated (blue): C - commercial/business, R - residential.

**RC 1.15** Lines 424-425: The air appears to become wetter aloft with PBL as shown in Figure 17.

**Reply**: Thanks for pointing out the incorrect statement - corrected as suggested:

At night, the air temperature in grid C stays warmer at lower altitudes but with cooler and wetter air aloft with PBL (Fig. 17).

RC 1.16 Lines 435-436: Kindly remove the redundant "explored".

**Reply**: Removed as suggested.

**Reviewer 2**

This manuscript described the structure and key physics of the coupled WRF-SUEWS systemand evaluated WRF-SUEWS at two UK sites and explored its application in modelling dynamics and impacts of anthropogenic heat emissions at the city scale. The topic is very interesting and has important implications in urban climate. However, there are major concerns which lead me to request a minor revision of this manuscript before publish.

**Reply**: We appreciate your recognition of our work.

**RC 2.1** The urban boundary layer fluctuates with weather scenario changes, especially for synoptic pattern. Synoptic patterns modulate local weather condition in boundary layer, e.g., SUEWS, boundary layer height, wind, RH and temperature, or even cloud. Therefore, the limitation and applicability of present coupled WRF-SUEWS system should be discussed, especially for some special synoptic patterns.

**Reply**: Related discussions have now been added in the "Concluding Remarks" section - please see reply to **RC1.13**.

**RC 2.2** In addition, for clear sky, the role of AHR and land use and their impacts on local climate in the London should be compared with other regions.

- Effects of anthropogenic heat release upon the urban climate in a Japanese megacity
- A High-Resolution Monitoring Approach of Canopy Urban Heat Island using Random Forest Model and Multi-platform Observations
- Simulating the Regional Impacts of Urbanization and Anthropogenic Heat Release on Climate across China
- Modulation of wintertime canopy Urban Heat Island (CUHI) intensity in Beijing by synoptic weather pattern in planetary boundary layer

**Reply**: We thank the reviewer for providing these useful references, of which the most relevant ones have now been discussed in Sect. 4.1 as follows:

Besides, the WRF-SUEWS predicted average  $Q_F$  values for London (i.e. ~ 18 W m-2) are comparable with those estimated in other mega-cities (e.g. peak values of ~ 50 W m-2 in Osaka City, Japan by Narumi et al. (2009); annual average of ~ 20 W m-2 in Beijing–Tianjin–Hebei Agglomeration by Feng et al. (2012); annual average of ~ 30 W m-2 in Beijing by Yang et al. (2022)).

**RC 2.3** Moreover, the observed boundary layer height should be described detailly and the accuracy should be pointed out for model validation.

**Reply**: This has now been added in the revised manuscript - please see reply to **RC1.6**.

**References**

- Feng, J.-M., Wang, Y.-L., Ma, Z.-G., and Liu, Y.-H.: Simulating the Regional Impacts of Urbanization and Anthropogenic Heat Release on Climate across China, J. Climate, 25, 7187–7203, https://doi.org/10.1175/jcli-d-11-00333.1, 2012.
- Järvi, L., Rannik, U., Kokkonen, T. V., Kurppa, M., Karppinen, A., Kouznetsov, R. D., Rantala, P., Vesala, T., and Wood, C. R.: Uncertainty of eddy covariance flux measurements over an urban area based on two towers, Atmospheric Measurement Techniques, 11, 5421–5438, https://doi.org/ 10.5194/amt-11-5421-2018, 2018.
- Jimenez, P. A., Hacker, J. P., Dudhia, J., Haupt, S. E., Ruiz-Arias, J. A., Gueymard, C. A., Thompson, G., Eidhammer, T., and Deng, A.: WRF-Solar: Description and Clear-Sky Assessment of an Augmented NWP Model for Solar Power Prediction, Bull. Amer. Meteorol. Soc., 97, 1249–1264, https://doi.org/10.1175/bams-d-14-00279.1, 2016.
- Kotthaus, S. and Grimmond, C.: Energy exchange in a dense urban environment Part I: Temporal variability of long-term observations in central London, Urban Clim., 10, 261–280, https://doi.org/10.1016/j.uclim.2013.10.002, 2014.
- Kotthaus, S. and Grimmond, C. S. B.: Atmospheric boundary-layer characteristics from ceilometer measurements. Part 1: A new method to track mixed layer height and classify clouds, Quart. J. Roy. Meteorol. Soc., 144, 1525–1538, https://doi.org/10.1002/qj.3299, 2018a.
- Kotthaus, S. and Grimmond, C. S. B.: Atmospheric boundary-layer characteristics from ceilometer measurements. Part 2: Application to London's urban boundary layer, Quart. J. Roy. Meteorol. Soc., 144, 1511–1524, https://doi.org/10.1002/qj.3298, 2018b.
- Kotthaus, S., O'Connor, E., Münkel, C., Charlton-Perez, C., Haeffelin, M., Gabey, A. M., and Grimmond, C. S. B.: Recommendations for processing atmospheric attenuated backscatter profiles from Vaisala CL31 ceilometers, Atmos. Meas. Tech., 9, 3769–3791, https://doi.org/ 10.5194/amt-9-3769-2016, 2016.
- Kotthaus, S., Bravo-Aranda, J. A., Collaud Coen, M., Guerrero-Rascado, J. L., Costa, M. J. a., Cimini, D., O'Connor, E. J., Hervo, M., Alados-Arboledas, L., Jiménez-Portaz, M., Mona, L.,

Ruffieux, D., Illingworth, A., and Haeffelin, M.: Atmospheric boundary layer height from groundbased remote sensing: A review of capabilities and limitations, Atmos. Meas. Tech., 16, 433–479, https://doi.org/10.5194/amt-16-433-2023, 2023.

- Lapo, K. E., Hinkelman, L. M., Sumargo, E., Hughes, M., and Lundquist, J. D.: A critical evaluation of modeled solar irradiance over California for hydrologic and land surface modeling, Journal of Geophysical Research: Atmospheres, 122, 299–317, https://doi.org/10.1002/2016jd025527, 2017.
- Mahrt, L., Thomas, C., Richardson, S., Seaman, N., Stauffer, D., and Zeeman, M.: Non-stationary Generation of Weak Turbulence for Very Stable and Weak-Wind Conditions, Bound.-Layer Meteorol., 147, 179–199, https://doi.org/10.1007/s10546-012-9782-x, 2012.
- Narumi, D., Kondo, A., and Shimoda, Y.: Effects of anthropogenic heat release upon the urban climate in a Japanese megacity, Environ. Res., 109, 421–431, https://doi.org/10.1016/j.envres. 2009.02.013, 2009.
- Yang, Y., Guo, M., Ren, G., Liu, S., Zong, L., Zhang, Y., Zheng, Z., Miao, Y., and Zhang, Y.: Modulation of Wintertime Canopy Urban Heat Island (CUHI) Intensity in Beijing by Synoptic Weather Pattern in Planetary Boundary Layer, Journal of Geophysical Research: Atmospheres, 127, https://doi.org/10.1029/2021jd035988, 2022.